# Direct Numerical Layout Generation for 3D Indoor Scene Synthesis via Spatial Reasoning

**Xingjian Ran**[1,2]  **Yixuan Li**[3]  **Linning Xu**[3]  **Mulin Yu**[2]  **Bo Dai**[1*]

[1]The University of Hong Kong  [2]Shanghai Artificial Intelligence Laboratory
[3]The Chinese University of Hong Kong
https://directlayout.github.io/

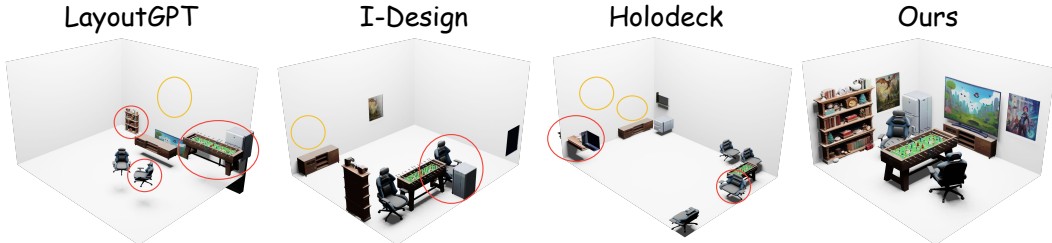

A foosball table is at the center of a game room, with two gaming chairs positioned on its long sides. A large TV sits on a wooden stand against the wall, behind the foosball table. To the left of the TV is a silver mini fridge. On the other wall, a tall bookshelf is filled with board games. Two posters are displayed on the walls, one next to the bookshelf and one next to the TV.

Figure 1: Our method synthesizes 3D indoor scenes from text descriptions via direct numerical layout generation, demonstrating strong performance in both instruction compliance and physical plausibility. In contrast, existing methods often suffer from issues related to inappropriate placement and size, as highlighted by the red circles. Furthermore, they struggle to identify all the entities in fine-grained user instruction resulting in object omission, indicated by the yellow circles. All methods share the same assets generated by 3D object generation method to ensure a fair comparison.

## Abstract

Realistic 3D indoor scene synthesis is vital for embodied AI and digital content creation. It can be naturally divided into two subtasks: object generation and layout generation. While recent generative models have significantly advanced object-level quality and controllability, layout generation remains challenging due to limited datasets. Existing methods either overfit to these datasets or rely on predefined constraints to optimize numerical layout that sacrifice flexibility. As a result, they fail to generate scenes that are both open-vocabulary and aligned with fine-grained user instructions. We introduce DirectLayout, a framework that directly generates numerical 3D layouts from text descriptions using generalizable spatial reasoning of large language models (LLMs). DirectLayout decomposes the generation into three stages: producing a Bird's-Eye View (BEV) layout, lifting it into 3D space, and refining object placements. To enable explicit spatial reasoning and help the model grasp basic principles of object placement, we employ Chain-of-Thought (CoT) Activation based on the 3D-Front dataset. Additionally, we design CoT-Grounded Generative Layout Reward to enhance generalization and spatial planning. During inference, DirectLayout addresses asset-layout mismatches via Iterative Asset-Layout Alignment through in-context learning. Extensive experiments demonstrate that DirectLayout achieves impressive semantic consistency, generalization and physical plausibility.

---

*Corresponding author.

39th Conference on Neural Information Processing Systems (NeurIPS 2025).

# 1 Introduction

High-fidelity, spatially coherent 3D indoor scenes are crucial for embodied AI, virtual reality, and film production. Particularly in advancing embodied AI, the training of agents for tasks like navigation and object manipulation relies heavily on realistic and diverse simulation environments. To address the complexity of 3D scene synthesis, the task can be divided into two stages: object generation and layout generation. Recent 3D generative models [28, 33, 35] have significantly advanced the quality and controllability of object generation. In contrast, layout generation remains underexplored, with key challenges including physical plausibility, semantic alignment, and diversity. Physical plausibility requires that objects are arranged in ways consistent with real-world physics, for instance, they should rest stably on surfaces and avoid unnatural overlaps. Semantic alignment means the layout must accurately reflect the user's textual instructions, such as placing a bed next to a window when requested. Diversity involves the ability to generate a wide variety of object types and room layouts, extending beyond typical bedrooms and living rooms to spaces like kitchens, offices, or playrooms.

Due to limitations in scale, diversity, and realism, existing 3D indoor layout datasets struggle to reflect the true distribution of real-world scenes. Directly training generative models on such datasets [16, 19, 30], often leads to models that simply memorize dataset-specific patterns, resulting in poor generalization to novel room and object types. To alleviate this data bottleneck, recent approaches [1, 8, 31] introduce manually defined spatial constraints, guiding the numerical layout generation process through intermediate representations such as scene graphs. These methods improve physical plausibility and reduce hallucinations. However, it inherently sacrifices flexibility and makes it challenging to accommodate fine-grained user instructions. This limitation stems from the reliance on predefined constraints, which struggle to capture the diversity and complexity of real-world layouts. Because object placements frequently depend on contextual, functional, and aesthetic factors, all of which are difficult to exhaustively specify with fixed rules.

To address the data challenge, we enable the model to learn generalizable spatial reasoning and directly generate numerical layouts without relying on predefined rules. Our pipeline uses large language models (LLMs) to generate 3D layout and object descriptions from textual input through decomposed processes. During generation, the core of spatial reasoning in LLMs lies in Chain-of-Thought (CoT), which comprises two key components. First, CoT Activation is introduced during training to enable structured reasoning steps, helping the model grasp fundamental spatial logic and forming a prior for spatial planning. This is achieved by supervising a four-step reasoning process that decomposes layout generation into subtasks, allowing the model to better understand and organize spatial information. Second, CoT-Grounded Generative Layout Reward provides a learning signal to improve generalizable spatial reasoning under limited data, by assessing the plausibility of object placements and their consistency with the CoT reasoning process. It introduces a dual-evaluator framework, where a vision-language model (VLM) captures high-level spatial and semantic violations, while a reasoning LLM identifies fine-grained numerical and logical inconsistencies, together offering structured, interpretable evaluation. After generation, we incorporate Iterative Asset-Layout Alignment to refine layout-object consistency and enhance scene realism based on spatial and semantic feedback provided by the aforementioned dual-evaluator. As shown in Figure 1, existing methods often suffer from inappropriate placements and object omission. In contrast, our approach generates 3D scenes with higher physical plausibility and better alignment with the user instruction.

Our contributions are summarized as follows:

- We propose a novel framework for 3D indoor scene synthesis that generates and refines layouts directly from text descriptions, bypassing the need for constrained optimization. This is achieved through a carefully designed task decomposition, which simplifies the process and improves overall efficiency.

- We introduce CoT reasoning into 3D scene synthesis, enhancing the spatial planning capabilities of the model. By connecting CoT reasoning with reinforcement learning through CoT-Grounded Generative Layout Reward, we significantly improve the model's reasoning accuracy and reduce hallucination. Additionally, we curate a CoT dataset for 3D indoor layouts to support this approach.

- Our DirectLayout outperforms existing methods in general 3D scene synthesis, demonstrating that direct generation of numerical layouts is highly effective. The generated layouts are more consistent with physical laws and show improved semantic consistency. Moreover, the direct generation of numerical layouts allows for better fine-grained control.

## 2 Related Work

### 2.1 3D Scene Synthesis Approaches

3D indoor scene synthesis methods vary in representation and generation strategy. Rule-based methods [4, 15, 22] provide structured layouts and are particularly suited for task-driven evaluation. In contrast, recent research has focused on learning-based models [14, 18, 27], which aim to generalize from data and facilitate the generation of more diverse and semantically rich scenes. ATISS [19] uses a permutation-equivariant transformer to generate object sets conditioned on a floor plan, supporting interactive editing. LayoutGPT [6] treats an LLM as a visual planner to predict layouts from text using CSS-like prompts, performing competitively in both 2D and 3D settings. Structured approaches like InstructScene [16] and AnyHome [8] incorporate scene graphs as intermediates to enhance control and semantic alignment. Holodeck [31] encodes relational constraints via LLM and solves a constraint optimization problem for object placement. LayoutVLM [23] combines visual grounding with differentiable optimization, refining object placements through relation-based objectives and initial numerical poses, while making deliberate trade-offs between flexibility and realism. [5] propose a hierarchical planning strategy for 3D scene generation using VLMs. Their method decomposes scenes into room, region, and object levels, using tree search to explore layout options and improve spatial plausibility. Previous approaches mainly focus on better modeling the distribution of indoor scene layouts, whereas our method leverages limited data to learn the underlying placement logic of indoor layouts.

### 2.2 Reasoning Augmentation in Vision

Recent advancements [9, 32, 34, 36] have utilized the general reasoning capabilities of LLMs and VLMs to tackle visual tasks. However, other works delve deeper into methods designed to explicitly enhance their reasoning abilities and performance within the visual domain. For instance, [3] prompts LLMs to interpret complex textual descriptions step-by-step, transforming them into structured 2D layouts that improve compositional accuracy, which are subsequently used to condition diffusion models. LLaVA-CoT [29] introduces a multi-stage reasoning framework which substantially boosts performance on complex visual question-answering benchmarks. SpatialCoT [17] enhances embodied task planning by aligning vision-language inputs with spatial coordinates and applying chain-of-thought spatial grounding, resulting in superior performance in navigation and manipulation tasks. [5] also enhance layout reasoning by discretizing the scene into symbolic grids and prompting VLMs to iteratively generate object placements, combining structured spatial reasoning with visual grounding. Furthermore, [11] integrates learned reward models, PARM and PARM++, into autoregressive image generation, facilitating stepwise verification that improves image quality and alignment. Drawing inspiration from these works, our method seeks to enhance layout reasoning through the use of explicit logical outputs and reward-based feedback.

## 3 Method

### 3.1 Problem Formulation

Given general textual descriptions and user-specified room dimensions, our goal is to synthesize physically and semantically plausible 3D indoor scenes. The synthesis process focuses on generating structured layouts, while 3D object assets are handled by existing generative methods. We represent the 2D top-down layout (BEV layout) as:

$$L_{\mathrm{BEV}} = \{(x_i, y_i, l_i, w_i, o_i)\}_{i=1}^{N}, \tag{1}$$

where $(x_i, y_i)$ is the center coordinate of object $i$, $(l_i, w_i)$ are its length and width, and $o_i$ is its orientation (rotation angle around the $z$-axis).

To extend this into 3D, we define the 3D layout as:

$$L_{\mathrm{3D}} = \{(x_i, y_i, z_i, l_i, w_i, h_i, o_i, p_i)\}_{i=1}^{N}, \tag{2}$$

where $z_i$ and $h_i$ denote the vertical center and height of object $i$, and $p_i$ is a text prompt used to guide 3D asset generation.

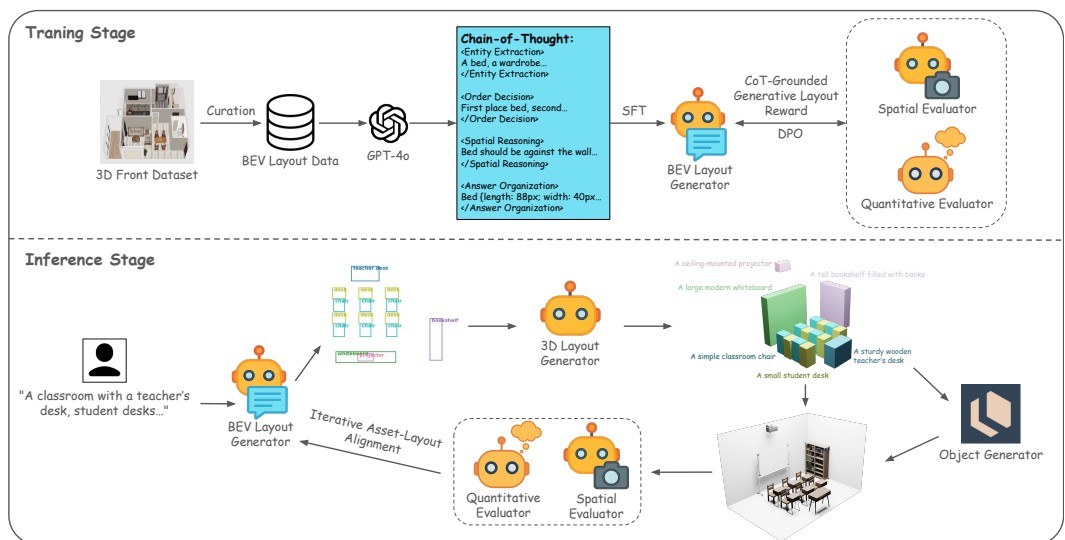

Figure 2: **Overview of our method. Training Stage:** BEV Layout Generator is first fine-tuned on BEV layouts curated from the 3D-Front dataset, guided by CoT annotations generated by GPT-4o. Subsequently, it is further optimized through DPO, leveraging CoT-Grounded Generative Layout Reward derived from Spatial Evaluator (VLM) and Quantitative Evaluator (reasoning LLM). **Inference Stage:** Given a text prompt, BEV Layout Generator produces a 2D layout, which is then lifted to a 3D layout by 3D Layout Generator. Iterative Asset-Layout Alignment refines the 3D scene by using the Spatial Evaluator and Quantitative Evaluator to provide feedback to the layout generators, ensuring consistency between the layout and generated 3D assets from an object generator.

## 3.2 Overview

Despite the powerful reasoning capabilities of recent foundation language models, we observe that they often lack spatial planning ability. To address this limitation, we decompose the scene synthesis task into three stages that are comparatively easier to handle, while preserving the overall spatial consistency, as illustrated in Figure 2. We first use a *BEV Layout Generator* to convert the input text into a 2D top-down layout. Section 3.3 details how we enhance the model's spatial reasoning through *CoT Activation* and *CoT-Grounded Generative Layout Reward*. The resulting BEV layout is then lifted into a 3D layout using the *3D Layout Generator*, which also assigns a descriptive text prompt to each object instance to facilitate asset generation (Section 3.4). Finally, we apply *Iterative Asset-Layout Alignment* to reconcile discrepancies between the predicted layout and available assets during inference, using feedback from evaluators and in-context learning (Section 3.5). All components in our pipeline, including generators and evaluators, are built upon LLMs or VLMs. Among them, only the BEV Layout Generator is explicitly fine-tuned for layout planning.

## 3.3 Enhancing Spatial Reasoning in BEV Layout Generation

### 3.3.1 CoT Activation

We leverage CoT reasoning to activate and guide the basic spatial planning capabilities of the BEV Layout Generator. Inspired by recent advancements in symbolic and mathematical reasoning with LLMs [10, 13, 26], we design a structured four-step CoT process. As illustrated in Figure 2, reasoning begins with **entity extraction**, where the model parses the input text to identify object categories and their quantities. This is followed by an **order decision** step, which establishes a semantically grounded placement sequence by prioritizing foundational objects. Next, **spatial reasoning** is performed to infer each object's location, size, and orientation, relying on common sense and relative spatial relationships. Finally, the **answer organization** step produces the BEV layout in a structured numerical format suitable for downstream use.

To supervise this reasoning process, we filter approximately 6,500 scenes from the 3D-Front dataset [7] following the pre-processing of ATISS [19], and use GPT-4o [12] to generate the CoT

annotations for ground-truth layouts, guided by carefully designed prompts provided in Appendix A.1. We then fine-tune BEV Layout Generator using supervised fine-tuning (SFT). While this improve layout quality and reasoning fidelity, we observe occasional hallucinations and inconsistency between reasoning and final placements, especially on unseen object or scene types.

### 3.3.2 CoT-Grounded Generative Layout Reward

Due to the lack of large-scale and diverse scene layout datasets, it remains challenging for fine-tuned BEV Layout Generator to generalize well to unseen or complex situations. To further improve generalizable reasoning in layout generation, we propose CoT-Grounded Generative Layout Reward, which assesses layout plausibility through a dual-evaluator framework: *Spatial Evaluator* and *Quantitative Evaluator*. This dual-evaluator framework benefits from the complementary strengths of a VLM and a reasoning LLM, leading to more robust layout assessment. The VLM serves as Spatial Evaluator, adept at capturing high-level spatial and semantic implausibility by leveraging both visual and textual context to evaluate the result and its consistency with CoT. In contrast, the reasoning LLM functions as Quantitative Evaluator, excelling at identifying fine-grained numerical and logical inconsistencies, particularly those that are more subtle and not easily discernible through visual cues alone. Moreover, instead of relying on absolute score prediction, which is difficult to the subjective and diverse nature of layout design, our reward focuses on detecting violations of physical and common sense constraints, aligning more closely with the pretraining priors of large models.

**Object-Level Criteria.** To provide a comprehensive evaluation of layout quality, we assess each generated layout at the object level using seven criteria divided between the two evaluators:

**Spatial Evaluator:**

- *Relative Alignment* ($C_1$): Measures the reasonableness of spatial relations with related objects.

- *Global Positioning* ($C_2$): Assesses if the position aligns with the room context and user prompt.

- *Consistency with CoT* ($C_3$): Ensures that object placements align with the spatial instructions inferred or explicitly stated in the CoT description.

**Quantitative Evaluator:**

- *Inter-object Distance* ($C_4$): Checks that spacing between objects supports accessibility and functionality. Overlaps may be acceptable if vertical stacking is plausible.

- *Size Proportion* ($C_5$): Evaluates whether object sizes are reasonable within the scene.

- *Orientation Validity* ($C_6$): Ensures that object orientations are semantically appropriate.

- *Quantity Alignment* ($C_7$): Validates whether the number of instances per object class aligns with expectations given the room's purpose.

Each criterion $C_k$ ($k \in \{1, 2, 3, 4, 5, 6\}$) is associated with a validity ratio $r_k$, which serves as the reward for the layout under that criterion. The ratio is defined as $r_k = O_k/N$, where $O_k$ denotes the number of objects that meet criterion $C_k$, and $N$ is the total number of objects in the layout. Thus, $r_k$ quantifies the degree to which the layout conforms to the corresponding requirement. For $C_7$, the quantity alignment criterion, we define:

$$
r_7 = \max \left( 0, \ 1 - \frac{\sum_{c=1}^{M} \left| n_c^{\text{actual}} - n_c^{\text{expected}} \right|}{\sum_{c=1}^{M} n_c^{\text{expected}}} \right) \tag{3}
$$

where $n_c^{\text{actual}}$ and $n_c^{\text{expected}}$ denote the actual and expected object count for class $c$, and $M$ is the number of object classes.

By integrating both evaluators, we construct a reward signal that captures both semantic and numeric fidelity of layouts, providing holistic feedback.

**Scene-Level Reward Aggregation.** To obtain a unified reward per sample, we apply the entropy weight method, as different scene descriptions impose uneven difficulty across criteria. Entropy captures the information content of each criterion in varying contexts, enabling more adaptive reward

aggregation. Given $T$ layout samples $\{S_1, ..., S_T\}$ for a prompt, and their criterion-wise validity ratios $\{r_k^{(j)}\}$ for sample $j$, the entropy of criterion $k$ is:

$$H_k = -\frac{1}{\ln T} \sum_{j=1}^{T} p_k^{(j)} \ln p_k^{(j)}, \quad \text{where } p_k^{(j)} = \frac{r_k^{(j)}}{\sum_{j=1}^{T} r_k^{(j)}} \tag{4}$$

The final reward $R^{(j)}$ for sample $S_j$ is computed as a weighted average of the criteria, where each weight is inversely related to the entropy $H_k$:

$$R^{(j)} = \sum_{k=1}^{7} \frac{(1 - H_k)\, r_k^{(j)}}{\sum_{k=1}^{7}(1 - H_k)}. \tag{5}$$

**Model Training.** We collect 200 diverse prompts with room dimensions using GPT-4o [12], featuring varying levels of detail (see Appendix A.3), and generate 30 BEV layout samples per prompt. Each sample is assigned a reward based on the above procedure. We construct the training set by sampling data pairs whose reward differences exceed a certain threshold (with the higher-scoring sample labeled as `chosen` and the lower-scoring one as `rejected`), and fine-tune BEV layout generator using the Direct Preference Optimization (DPO) [21]. More training details can be found in Appendix A.2.

### 3.4   Layout Lifting and Scene Synthesis

Given the predicted BEV layout, we employ 3D Layout Generator to infer missing vertical attributes such as object heights and vertical positions, guided by the scene description and structured BEV layout. It leverages the LLM's strong prior knowledge of object sizes and height relationships (e.g., beds are lower than tables, lamps above desks), which are generally common sense, enabling effective zero-shot inference. In addition to completing vertical attributes, it generates a descriptive prompt $p_i$ for each object, capturing category, shape, and style to support semantically aligned 3D asset generation.

To synthesize the final 3D scene, we generate assets from the prompts using an existing high-fidelity 3D object generation model [28]. Each asset is resized according to $(l_i, w_i, h_i)$ and positioned using $(x_i, y_i, z_i)$ and $o_i$, thus completing the full scene assembly pipeline.

### 3.5   Iterative Asset-Layout Alignment

Although the generated prompts contain detailed textual descriptions, the generators lack direct access to visual information about 3D assets, often leading to inconsistencies between predicted layouts and generated objects. Specifically, the orientation of generated objects can be inconsistent, and instances of the same category may differ significantly in shape, such as a bathtub filled with water compared to an empty one, leading to uncertainties in the height of related objects (see Figure 6 for an example).

To address this, we propose Iterative Asset-Layout Alignment. After rendering the composed 3D scene, we re-evaluate it using the same Spatial Evaluator and Quantitative Evaluator introduced in Section 3.3, based on a shared set of object-level criteria $C_{k}{}_{k=1}^{7}$. For each violated criterion, the evaluators provide object-specific suggestions to adjust size, position, or orientation. These are then fed back into the layout generators as contextual input for iterative refinement. This process repeats until either no further revisions are proposed by the evaluators or a maximum number of refinement steps is reached. Let $L_{3D}^{(t)}$ denote the 3D layout at iteration $t$, and let $\mathcal{F}^{(t)}$ represent the evaluator feedback at iteration $t$, which consists of a set of suggestions or corrections based on the violation of spatial or semantic criteria. The updated layout is defined as:

$$L_{3D}^{(t+1)} = \texttt{Update}(L_{3D}^{(t)}, \mathcal{F}^{(t)}), \tag{6}$$

where `Update` applies layout modifications suggested by the evaluators. Through this looped alignment mechanism, our system is able to dynamically correct mismatches between layout and asset geometry, resulting in coherent and realistic 3D scenes.

| LayoutGPT | I-Design | Holodeck | Ours |

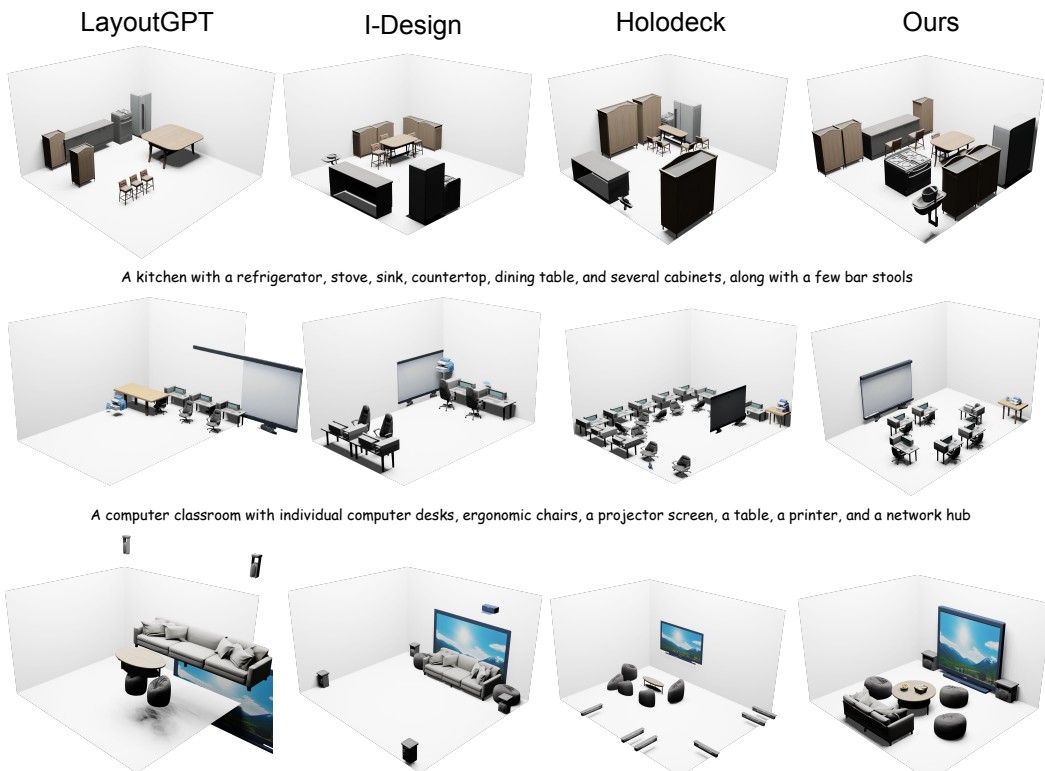

A kitchen with a refrigerator, stove, sink, countertop, dining table, and several cabinets, along with a few bar stools

A computer classroom with individual computer desks, ergonomic chairs, a projector screen, a table, a printer, and a network hub

In a home theater, a large screen is positioned against a wall, flanked by two speakers. In front of it, a round coffee table holds popcorn and potato chips. A sofa sits behind the coffee table, and several bean bags are arranged around it.

A wooden table holds a laptop on the front left corner, with a glass of colored pencils placed to its right. At the center back of the table sits a blue vase filled with flowers, and in front of the vase lie a pair of black-framed glasses. On the left side of the table is a white coffee cup, while a wooden chair is positioned in front of the table.

Figure 3: **Qualitative comparisons with scene synthesis methods.** We compared our generated scenes with existing methods across various scene types and coarse-to-fine prompt granularities. Our results demonstrate a better alignment with the text descriptions across different prompt granularities and scene types.

## 4 Experiments

### 4.1 Experimental Setup

We evaluate our approach on a set of 15 indoor scene categories, each associated with three distinct prompts and room sizes. These test indoor scenes are automatically generated and manually verified to ensure consistency across categories. To assess the effectiveness and reproducibility of our method, we implement two distinct system configurations. In the **Closed-Source Enhanced** setting, we utilize GPT-4o [12] as both 3D Layout Generator and Spatial Evaluator, while o1 [13] serves as Quantitative Evaluator. In contrast, the **Open-Source Only** configuration offers a fully reproducible alternative by employing Qwen2.5-72B [24] as 3D Layout Generator, Qwen2.5-VL-72B [25] as Spatial Evaluator, and QwQ-32B [26] as Quantitative Evaluator.

In both configurations, we use a fine-tuned Qwen2.5-32B as BEV Layout Generator. We compare our method against several recent baselines in text-to-layout generation, including LayoutGPT [6],

Table 1: **Comparison with existing methods.** Our approach outperforms baselines in both physical plausibility and semantic alignment on general indoor scenes.

| Method | Out of Bound ↓ | Collision ↓ | Pos. ↑ | Rot. ↑ | PSA ↑ | CLIP Score ↑ |
|---|---|---|---|---|---|---|
| Holodeck | 36.33 | 8.32 | 51.56 | 50.18 | 49.78 | 25.21 |
| I-Design | 16.00 | 18.85 | 13.47 | 13.82 | 29.33 | 14.09 |
| LayoutGPT | 33.91 | 15.89 | 57.24 | 51.31 | 55.11 | 21.65 |
| Ours w/ QwQ BEV | 18.85 | 12.50 | 44.33 | 43.64 | 47.11 | 23.83 |
| Ours (open-source) | 9.20 | 8.45 | 70.73 | 67.51 | 73.78 | 26.47 |
| Ours (full) | 8.74 | 6.31 | 72.04 | 69.87 | 75.56 | 27.43 |

Holodeck [31], and I-Design [2]. Additionally, we construct a strong baseline by replacing our specialized BEV Layout Generator with a general reasoning LLM (QwQ-32B) to examine the effect of inductive spatial bias.

**Evaluation Metrics:** To evaluate *physical plausibility*, we employ two metrics: Out-of-Bound Rate and Collision Rate. The Out-of-Bound Rate is calculated as the proportion of objects that extend beyond the scene boundaries relative to the total number of objects. Similarly, the Collision Rate represents the proportion of objects that collide with other objects relative to the total number of objects. Following LayoutVLM [23], we assess *semantic alignment* and overall quality using Positional Coherency (Pos.), Rotational Coherency (Rot.), and the Physically-Grounded Semantic Alignment Score (PSA). In LayoutVLM, these LLM-based metrics, evaluated using GPT-4o, have been shown to correlate with human judgments. Additionally, we report the CLIP Score [20] to quantify the consistency between the generated layout and the input text prompt.

## 4.2 Comparison with Baselines

We conduct a series of comparative evaluations aiming to verify the advantages of our approach over baselines. These comparisons reveal the extent to which our model improves physical feasibility, semantic fidelity, and user-controllable scene synthesis.

**Comparison with Existing Methods.** Table 1 and Figure 3 summarize the comparative results across key metrics and visual outputs. LayoutGPT directly predicts object placements using in-context examples. However, the absence of logical training leads to outputs that violate physical constraints and deviate from user instructions. Its generated layouts often lack structural coherence and semantic fidelity. Although LayoutGPT employs direct numerical layout generation, its limited spatial reasoning hampers its ability to translate detailed descriptions into coherent layouts.

While I-Design leverages cooperative reasoning among multiple agents to coordinate layouts, it suffers from limited robustness when generating layout, particularly due to failures in agent communication or layout grounding. Although it produces physically plausible arrangements, the generated objects tend to cluster locally, lacking global semantic awareness. This is partly because I-Design extracts symbolic constraints before layout generation, which improves physical plausibility but limits the model's flexibility in handling complex and fine-grained spatial relationships often found in natural language, such as "a vase in the center of the table, with a pair of glasses in front of it." Its reliance on predefined constraints ultimately restricts controllability and semantic fidelity.

Holodeck adopts a search-based strategy that effectively avoids object collisions, but similarly falls short in capturing prompt semantics, often generating layouts with limited object diversity and local focus. Like I-Design, Holodeck also relies on symbolic constraint, which enhances physical validity but weakens its ability to express detailed spatial instructions described in natural language.

Conversely, our open-source and full model enable both accurate physical constraint satisfaction and faithful semantic grounding. By integrating spatial reasoning directly into the numerical layout generation process without relying on symbolic constraints, our method maintains high physical plausibility and semantic alignment across all levels of prompt granularity—from coarse scene and object types to fine-grained spatial instructions—as evidenced in Figure 3.

**Comparison with Reasoning LLM.** To isolate the contribution of our specialized BEV Layout Generator, we replace it with a general-purpose reasoning model, QwQ-32B, while keeping the rest of our pipeline unchanged. This variant—referred to as *Ours w/ QwQ BEV* in Table 1—serves as an ablation baseline to evaluate the role of spatial inductive bias. Thanks to its strong capabilities in numerical computation and local logical inference, QwQ-32B handles basic geometric constraints

Table 2: **Ablation study.** We evaluate the effect of task decomposition and reward granularity.

| Method | Out of Bound ↓ | Collision ↓ | Pos. ↑ | Rot. ↑ | PSA ↑ | CLIP Score ↑ |
|---|---|---|---|---|---|---|
| w/o decomp | 21.42 | 20.52 | 67.13 | 65.47 | 71.56 | 26.05 |
| w/o reward | 26.04 | 16.77 | 64.47 | 57.18 | 68.82 | 25.25 |
| simple reward | 13.64 | 11.89 | 67.95 | 64.37 | 70.76 | 25.82 |
| Ours (full) | 8.74 | 6.31 | 72.04 | 69.87 | 75.56 | 27.43 |

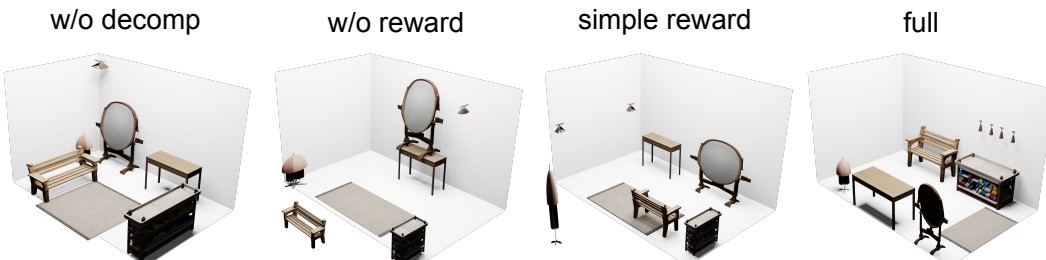

A hallway with a shoe rack, coat hooks, mirror, bench, umbrella stand, table, and a rug.

Figure 4: **Ablation Study Results.** The experiment validates the effectiveness of task decomposition and proposed CoT-Grounded Generative Layout Reward.

relatively well, maintaining acceptable rates of boundary and collision violations. However, the model often fails to preserve the holistic semantic context of a scene, leading to degraded performance in positional coherency, rotational alignment, and overall semantic alignment. This result suggests that general-purpose models, while strong at localized inference, lack the domain-specific priors required for globally consistent layout synthesis. These findings, as reflected in Table 1, highlight the advantage of incorporating dedicated spatial reasoning: they not only enforce physical feasibility but also enhance high-level understanding of spatial semantics.

## 4.3 Ablation Study

To gain a deeper understanding of the contribution of different components in our design, we conducted a series of ablation experiments focusing on the reward formulation and task decomposition. These experiments were designed to isolate the impact of each factor on the overall performance.

**CoT-Grounded Generative Layout Reward.** We investigate the impact of different reward configurations on the spatial plausibility of the generated layouts. As shown in Table 2 and illustrated in Figure 4, when BEV Layout Generator is trained solely with supervised learning, without any reinforcement signal (*w/o reward*), the physical plausibility significantly deteriorated. This is evident in the substantially higher rates of objects being placed outside the scene boundaries and colliding with each other. Introducing a simple reward, where a VLM provided a single overall score for the layout quality (*simple reward*; see Appendix A.1 for details), leads to some improvement compared to the no-reward scenario. However, this approach still fall short of our full method's performance. These results suggest that relying on high-level semantic feedback from a general foundation model is not sufficient for effectively guiding the learning of physically realistic layouts. In contrast, our method, which utilizes fine-grained, spatially informed rewards, achieved considerably better results across all evaluation metrics presented in Table 2 and visualized in Figure 4. This underscores the critical role of structured spatial supervision in generating plausible indoor scenes.

**Task Decomposition.** We evaluate the significance of our architectural modularity by comparing our full model with a variant where 3D layout generation and refinement were performed by a single model, without the intermediate 2D-to-3D reasoning step (*w/o decomp*). The results, detailed in Table 2 and qualitatively shown in Figure 4, indicate that this approach leads to a decline in physical plausibility and slightly weaker global semantic alignment. These results suggest that our decomposition strategy plays a crucial role in enabling more accurate spatial reasoning. By separating the 2D planning stage from the 3D realization, the model appears to be better equipped to handle the inherent complexity and variability of indoor environments.

Table 3: **Human evaluation results across different prompt granularities.** Higher scores indicate better semantic compliance (SemComp), physical plausibility (PhyPlaus), and layout rationality (LayRat).

| Method | Level | SemComp ↑ | PhyPlaus ↑ | LayRat ↑ |
|---|---|---|---|---|
| LayoutGPT | coarse | 3.72 | 1.84 | 1.72 |
| | medium | 2.88 | 1.64 | 1.76 |
| | fine | 3.00 | 2.32 | 2.04 |
| I-Design | coarse | 3.72 | 2.72 | 2.44 |
| | medium | 3.08 | 3.36 | 2.48 |
| | fine | 3.32 | 3.72 | 2.72 |
| Holodeck | coarse | 4.12 | 3.64 | 2.84 |
| | medium | 2.80 | 2.52 | 2.48 |
| | fine | 2.40 | 2.60 | 2.04 |
| Ours | coarse | 4.56 | 4.64 | 4.52 |
| | medium | 4.60 | 4.68 | 4.64 |
| | fine | 4.84 | 4.72 | 4.80 |

## 4.4 User Study

To further validate our approach, we conduct a user study with 25 compensated volunteers, all with relevant professional expertise. Participants are asked to evaluate scene layouts generated by different methods. The layouts are randomly ordered and anonymized. Each participant scores them according to three criteria: *Semantic Compliance* (faithfulness to the textual description), *Layout Rationality* (plausibility of object placement), and *Physical Plausibility* (realism of object support and absence of penetration). Here, *coarse*, *medium*, and *fine* indicate the granularity of the input prompt, with more detailed descriptions at finer levels (see Appendix A.3 for examples).

As shown in Table 3, our method consistently outperforms all baselines across coarse, medium, and fine prompt levels, with especially large margins at the fine-grained level. This demonstrates our model's strong ability to capture detailed semantic instructions while ensuring physically plausible placements. Compared to prior methods that rely on symbolic constraints or search heuristics, our approach maintains both global semantic consistency and local physical validity.

## 5 Conclusion

In this paper, we present DirectLayout, a novel framework for generating 3D indoor scenes from text descriptions via direct numerical layout generation. It follows a three-stage process including BEV layout generation, 3D lifting, and iterative refinement, enhanced by CoT Activation and CoT-Grounded Generative Layout Reward to improve generalizable spatial reasoning. Extensive experiments show that DirectLayout achieves state-of-the-art performance in generating general 3D scenes that are physically plausible, semantically coherent, and faithful to user prompts. We demonstrate that generating layouts through direct numerical generation allows for a higher upper bound in handling more detailed and complex user inputs and scene arrangements.

**Limitations.** Despite these advancements, the task decomposition and Iterative Asset-Layout Alignment, while effective, introduce additional inference time, which limits real-time interaction. Additionally, the complexity of the generated scenes is constrained by the capacity of the base model as well as the scale and diversity of scenes in the training dataset, making it challenging to generate a large number of small objects. Moreover, although representing scenes with a intermediate BEV layout generally yields better results than directly generating full 3D layouts, BEV layout inherently reduces layout clarity, when multiple objects are stacked along the vertical axis.

## Acknowledgments and Disclosure of Funding

The authors would like to thank the support from the HKU Startup Fund. This work is also partially funded by the National Key R&D Program of China (2022ZD0160201) and the Shanghai Artificial Intelligence Laboratory.

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

# A Implementation Details

## A.1 Prompts

In this section, we provide prompts used for training and testing the model.

**CoT Data Generation.** In order for the model to explicitly output logical thinking, we use the following prompt to generate CoT data for BEV Layout Generator (In ablation experiments, we use the same prompt to generate 3D layout CoT data, simply replacing the BEV layout format with the 3D layout format):

```
Given a BEV layout for a scene, first output a short prompt
    summarizing the scene, and then write a logical thought
    process when planning this layout, modeled after the
    following chain of thought example. The layout follows the
    CSS style, where each line starts with the object description
     and is followed by its absolute position. Formally, each
    line should be like "object {length: ?px; width: ?px;
    center_x: ?px; center_y: ?px; orientation: ? degrees;}". You
    can simply interpret the length and width as the dimensions
    of the object itself, with center_x and center_y indicating
    translation (or the center point of the object) and
    orientation indicating rotation. Note that the 0 degrees
    representation is aligned with the direction of the positive
    half-axis of the axes where "width" and "center_y" are
    located. The image is {max_length}px long and {max_width}px
    wide. Therefore, all bounding boxes should try not to exceed
    256px after rotation.

Below are the steps of the chain of thought:
1. Extract from the text description which objects should be
    placed in the scene and how many of each of these are
    specifically needed.
2. List the order in which to place each type of object,
    generally starting with the large and major objects, then
    moving on to the decorative and minor objects associated with
     them.
3. Place each object in the order in 2. Each object is placed by
     first giving the dimension of the object, then the rotation,
     and finally calculating the center point coordinates based
    on where it should be in the scene. The process should take
    into account the position of the object in the scene, its
    relative position to the placed objects, and the constraints
    between the objects. In this step, you need to not only place
     the object, but also give a detailed reason for placing it
    so.
4. Organize the answers given in 3. to produce a final output
    that meets the format requirements.

Here is how the json format output should look:
{
  "prompt": "(The prompt you generate)",
  "response": {
    "Entity Extraction": "(Explanation of the objects extracted
        from the prompt)",
    "Order Decision": "(Explanation of the order in which the
        objects should be placed, usually starting with major or
        large objects in the scene)",
    "Spatial Reasoning": "(Explanation of the dimensions,
        rotation, and position (i.e. center point) of each object
        , reasoning about each object should be as detailed as
```

```
              possible and take into account the scene and other
              objects)",
          "Answer Organization": "(Final output in the required format
              , using lines like "object {length: ?px; width: ?px;
              center_x: ?px; center_y: ?px; orientation: ? degrees;}")"
      }
  }
```

**Layout Lifting.** To complete each object's 3D pose and textual description, we use the following prompt as input to 3D Layout Generator:

```
Given a sentence prompt that will be used to generate a scene
    and the BEV layout of this scene, lifting a 2D layout to a 3D
     layout (i.e., predicting the range of heights an object
    occupies in space) and designing a prompt for each object for
     object asset generation. The generated layout should follow
    the CSS style, where each line starts with the object
    description and is followed by its absolute position.
    Formally, each line should be like "object {length: ?px;
    width: ?px; height: ?px; center_x: ?px; center_y: ?px;
    center_z: ?px; orientation: ? degrees;}". The given BEV
    layout contains information other than height and center_z,
    so the only information you need to add is about the objects
    in the z-axis. Be careful not to let objects that don't make
    sense appear overlapping. For example, chairs can go under
    tables, so they can overlap. And a lamp must go on top of a
    table, so they can't overlap. The space is {max_length}px
    long, {max_width}px wide, and 160px high. Therefore, the
    height of the bounding box should not exceed {max_height}px,
    i.e. center_z +/- height/2 needs to be between 0 and {
    max_height}. At the same time, for each object, prompt for
    objects should be in one sentence of natural language,
    describing its category, shape, and style. Finally give a
    list in the same order as the objects in the layout, like [
    obj1_prompt, obj2_prompt, ...].

Here is how the json format output should look:
{
  "3D_layout": [
    "bed {length: 88px; width: 40px; height: 36px; center_x: 120
        px; center_y: 60px; center_z: 18px; orientation: 0
        degrees;}",
    ...
  ],
  "object_prompts": [
    "A modern single bed with a rectangular frame and a wooden
        headboard.",
    ...
  ]
}

Prompt:
{text_description}
BEV layout:
{bev_layout}
```

**Quantitative Evaluator Feedback.** To obtain detailed, numerical feedback, we use the following prompt as input to Spatial Evaluator:

```
You are an expert assistant who is well versed in indoor scene
    layout design. As an impartial judge, you are asked to make
    an in-depth evaluation of a 2D scene layout (BEV layout),
    which is given by the bounding box of each object in the top
    view and labeled with the object category near the bounding
    box. You can refer to the metadata of the layout to make a
    judgment. You need to judge whether each type of object in
    the input BEV scene layout is in a reasonable position on
    each of the following four dimensions:
1. Distance between objects: Evaluate whether objects are spaced
     appropriately to ensure functionality and accessibility. For
     example, in organized environments such as classrooms or
    offices, furniture should be arranged with sufficient
    separation to allow for easy movement. Desks in a classroom
    should be spaced out so students can walk between them
    comfortably, and sofas or desks in offices should not overlap
    . Note that in a top-down view, some overlap in bounding
    boxes may be physically reasonable when objects are
    vertically stacked, such as a ceiling lamp above a bed, or a
    computer placed on a desk. In such cases, evaluating bounding
     box intersections alongside height context is essential.
2. Quantitative alignment of objects: Assess whether the
    quantities of related objects are consistent with functional
    expectations. For instance, in an office setting, the number
    of desks and chairs should correspond-each desk should
    typically be paired with one chair.
3. Size proportion of objects within the scene: Check whether
    the relative sizes of objects make sense within the spatial
    layout. For example, equipment such as computers in an office
     or lab instruments in a laboratory should be smaller than
    the tables or workbenches they are placed on. Unnatural size
    ratios, like a monitor larger than its desk, can suggest
    spatial implausibility.
4. Orientation of objects: Verify that objects are oriented
    appropriately for their intended use within the layout. For
    example, chairs in a classroom should face the blackboard,
    and computer monitors in an office should be directed toward
    the associated seating positions.

Please note that your choice must be based on a thorough
    understanding, analysis and evaluation of the image and the
    problem. After the explanation of each dimension, answer the
    final evaluation. Finally, return your judgment in a legal
    JSON format, evaluating Yes if the location of a particular
    object is considered reasonable in this dimension, and No if
    it is not. The json format and field definitions are as
    follows:
{
"object_class_name": ["Yes" or "No", "Yes" or "No", "Yes" or "No
    ", "Yes" or "No"] (four judgments correspond to the previous
    four dimensions)
...
}
scene description: {scene_description}
max_length: {max_length} px (horizontal axis)
max_width: {max_width} px (vertical axis)
BEV layout:
{bev_layout}
{metadata}
```

**Spatial Evaluator Feedback.** To obtain high-level, semantic feedback, we use the following prompt as input to Spatial Evaluator:

```
You are an expert assistant who is well versed in indoor scene
    layout design. As an impartial judge, you are asked to make
    an in-depth evaluation of the 2D scene layout (BEV layout),
    which is given by the bounding box of each object in the top
    view and labeled with the object category near the bounding
    box. You can refer to both the picture and the layout
    metadata for judging. You need to judge whether the
    information about each kind of object in the input BEV scene
    layout is reasonable in each of the following three
    dimensions:
1. Spatial alignment between objects: For example, the podium
    and projector in a classroom should be centrally aligned
    either horizontally or vertically, and lockers in locker
    rooms should be arranged in an orderly grid.
2. Position of the objects within the layout: Consider whether
    each object's position is appropriate for the overall layout
    context. For example, in a single office the desk should be
    near the center, and in a bank the counter should be near a
    wall.
3. Consistency with Chain-of-Thought Descriptions: The physical
    placements of objects should align with any provided textual
    chain-of-thought descriptions. For example, the chain of
    thought mentions placing the whiteboard next to the wall, but
     in the picture and digital layout the whiteboard is in the
    middle of the scene, which is unreasonable.

Please note that your choices must be based on a thorough
    understanding, analysis, and evaluation of the image and
    problem. After the explanation of each dimension, answer the
    final evaluation. Return your judgment at the end of each
    dimension in a legal JSON format, evaluating Yes if the
    placement of a particular object is considered reasonable in
    that dimension, and No if not. The json format and field
    definitions are as follows:
{
"object_class_name": ["Yes" or "No", "Yes" or "No", "Yes" or "No
    "] (three judgments correspond to the previous three
    dimensions)
...
}
scene description: {scene_description}
max_length: {max_length} px (horizontal axis in BEV)
max_width: {max_width} px (vertical axis in BEV)
BEV layout:
{bev_layout}
chain of thought:
{CoT}
```

The above two prompts are used to provide input to the evaluators during training, in order to compute
CoT-Grounded Generative Layout Reward. During inference, we use similar prompts as input to
the evaluators to perform Iterative Asset-Layout Alignment, with the only change being that the
output format is adapted to provide object-level suggestions (while keeping the evaluation criteria
unchanged).

**Scene Description Generation Prompt.** To construct diverse training and evaluation data, we use
the following prompt to instruct a language model to generate indoor scene descriptions at three
levels of granularity across multiple scene categories:

```
You are asked to generate indoor scene descriptions at three
    levels of granularity: coarse, medium, and fine-grained.
    Please follow the instructions carefully.
There are three types of granularity:
1. Coarse: List the main objects in the room without mentioning
    where they are.
    Example: "A home gym with a treadmill, yoga mat, dumbbell
        rack, water dispenser, and a large mirror."
2. Medium: Describe the approximate spatial relationships
    between major object groups.
    Example: "In a playroom, a toy shelf stands against the right
        wall, a bean bag lies in the corner near the window, and
        a round play mat is placed in the center."
3. Fine-grained: Provide precise, detailed spatial arrangements
    among individual objects.
    Example: "A small square table is placed in the center of the
        room. On the front right corner of the table sits a red
        toolbox, with a measuring tape coiled beside it. A yellow
        stool is tucked in on the left side, and a desk lamp
        stands at the rear center of the table."

You will generate scene descriptions for {num_scene_types}
    different indoor scene categories, excluding common
    categories such as bedroom, living room, dining room, and
    study.
For each scene category, generate:
{num_coarse_per_type} descriptions at coarse granularity,
{num_medium_per_type} at medium granularity,
{num_fine_per_type} at fine-grained granularity.
Each scene description should be accompanied by room dimensions:
    length, width, and height (each must be an integer <= 256).

Output the results in a strict json format as follows:
[
  {
    "scene_type": "laundry room",
    "granularity": "coarse",
    "description": "A laundry room with a washing machine, dryer
        , laundry basket, shelves, and detergent bottles.",
    "room_size": {
      "length": 256,
      "width": 171,
      "height": 240
    }
  },
  ...
]
```

**Simple Reward.** To verify the effectiveness of CoT-Grounded Generative Layout Reward, we use GPT-4o to give a overall score for BEV layout.

```
You are an expert assistant who is well versed in indoor scene
    layout design. As an impartial judge, you are asked to make
    an in-depth evaluation of the 2D scene layout (BEV layout),
    which is given by the bounding box of each object in the top
    view and labeled with the object category near the bounding
    box. You can refer to both the picture and the layout
    metadata for judging. Please score the scene based on
```

```
    physical plausibility, semantic consistency, and degree of
    instruction compliance.

Return your answer in legal JSON format. The format and field
    definitions are as follows:
{
"score": 1-100
}
scene description: {scene_description}
max_length: {max_length} px (horizontal axis in BEV)
max_width: {max_width} px (vertical axis in BEV)
BEV layout:
{bev_layout}
chain of thought:
{CoT}
```

## A.2 Training Details

We adopt a two-stage training pipeline to fine-tune the Qwen-2.5-32B. In the first stage, SFT is conducted on a generated CoT-augmented dataset containing approximately 6,500 CoT data of BEV layout. Low-Rank Adaptation (LoRA) with a rank of 8 is applied to all target modules. The model is trained for 3 epochs using a batch size of 1 and a gradient accumulation step of 8, with a learning rate of $1 \times 10^{-4}$. A cosine learning rate scheduler with 10% warm-up is employed, and training is performed using bfloat16 precision. The SFT stage is conducted on a single NVIDIA A800 GPU and takes approximately 8 hours to complete.

In the second stage, DPO is employed to further enhance spatial reasoning via CoT-Grounded Generative Layout Reward. The DPO dataset contains around 12,000 preference pairs, sampled such that the reward difference between the preferred and rejected generations exceeds 20. The same LoRA configuration and optimization hyperparameters are used as in the SFT stage. The preference loss function is set to `sigmoid` with $\beta = 0.1$. This stage is trained on 8 NVIDIA A800 GPUs for approximately 11 hours.

## A.3 Indoor Scene Description Generation Details

To support both training and evaluation, we generate a diverse set of textual indoor scene descriptions at three levels of granularity using GPT-4o following the prompt described in Appendix A.1. These levels are designed to test and guide the model's ability to interpret spatial semantics:

1. **Coarse**: Lists the key objects in the scene without specifying spatial relationships.
   *Example:* "A kitchen with a refrigerator, stove, sink, countertop, dining table, and several cabinets, along with a few bar stools."

2. **Medium**: Describes basic spatial layout and object grouping.
   *Example:* "In a home theater, a large screen is positioned against a wall, flanked by two speakers. In front of it, a round coffee table holds popcorn and potato chips. A sofa sits behind the coffee table, and several bean bags are arranged around it."

3. **Fine-grained**: Specifies detailed relative positions among objects.
   *Example:* "A wooden table holds a laptop on the front left corner, with a glass of colored pencils placed to its right. At the center back of the table sits a blue vase filled with flowers, and in front of the vase lie a pair of black-framed glasses. On the left side of the table is a white coffee cup, while a wooden chair is positioned in front of the table."

**Training.** For DPO training, we generate 200 scene descriptions across 40 indoor scene categories that are not present in the 3D-Front dataset (e.g., excluding bedrooms, living rooms, dining rooms, and studies). Each category includes 5 descriptions and room sizes, with the three granularity levels distributed in a 2:2:1 ratio (coarse:medium:fine-grained). These descriptions are directly used as prompts to generate layout samples for DPO training.

**Evaluation.** For evaluation, we construct a separate set of 45 test cases across 15 indoor scene categories, also distinct from those in 3D-Front. Each category is associated with three manually crafted and verified descriptions (one per granularity level), along with a specific room size. All descriptions are manually reviewed to ensure logical and spatial consistency. Minor corrections are made where necessary—for example, adding missing chairs in office scenes—to ensure a realistic and challenging testbed for layout generation.

# B   More Generated Results

In Figure 5, we present additional qualitative results in a variety of scene types, including garages, classrooms, bathrooms, and wine cellars. Our method consistently achieves strong physical plausibility and semantic alignment with language instructions, allowing users to exert fine-grained control over the generated content.

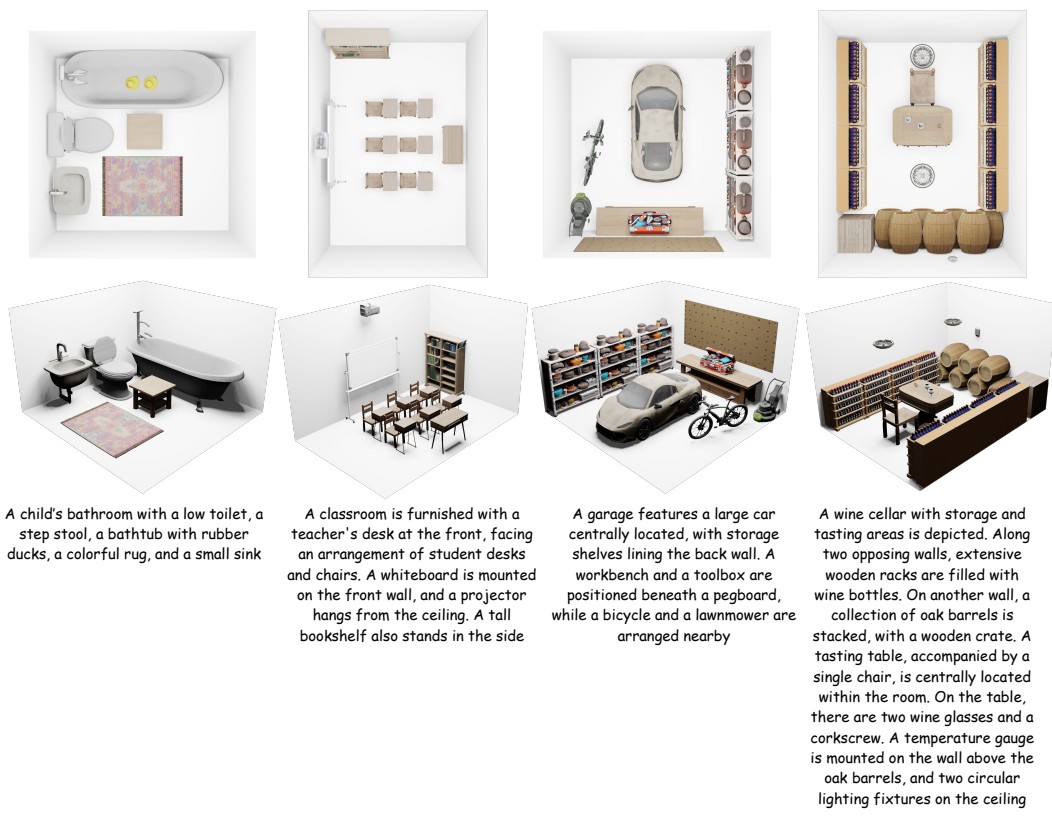

A child's bathroom with a low toilet, a step stool, a bathtub with rubber ducks, a colorful rug, and a small sink

A classroom is furnished with a teacher's desk at the front, facing an arrangement of student desks and chairs. A whiteboard is mounted on the front wall, and a projector hangs from the ceiling. A tall bookshelf also stands in the side

A garage features a large car centrally located, with storage shelves lining the back wall. A workbench and a toolbox are positioned beneath a pegboard, while a bicycle and a lawnmower are arranged nearby

A wine cellar with storage and tasting areas is depicted. Along two opposing walls, extensive wooden racks are filled with wine bottles. On another wall, a collection of oak barrels is stacked, with a wooden crate. A tasting table, accompanied by a single chair, is centrally located within the room. On the table, there are two wine glasses and a corkscrew. A temperature gauge is mounted on the wall above the oak barrels, and two circular lighting fixtures on the ceiling

Figure 5: More generated results based on language instructions with different granularities.

# C   Illustration of Iterative Asset-Layout Alignment

Figure 6 provides a visual illustration of Iterative Asset-Layout Alignment process described in Section 3.5. This figure highlights the feedback loop between scene generation and evaluation, where each rendered 3D scene undergoes spatial and semantic evaluation to identify inconsistencies. Based on the evaluator's feedback, targeted corrections are applied to the layout, including object size, position, and orientation. The updated layout is then used to regenerate the scene. This iterative refinement continues until the evaluators no longer detect violations or a preset iteration limit is reached.

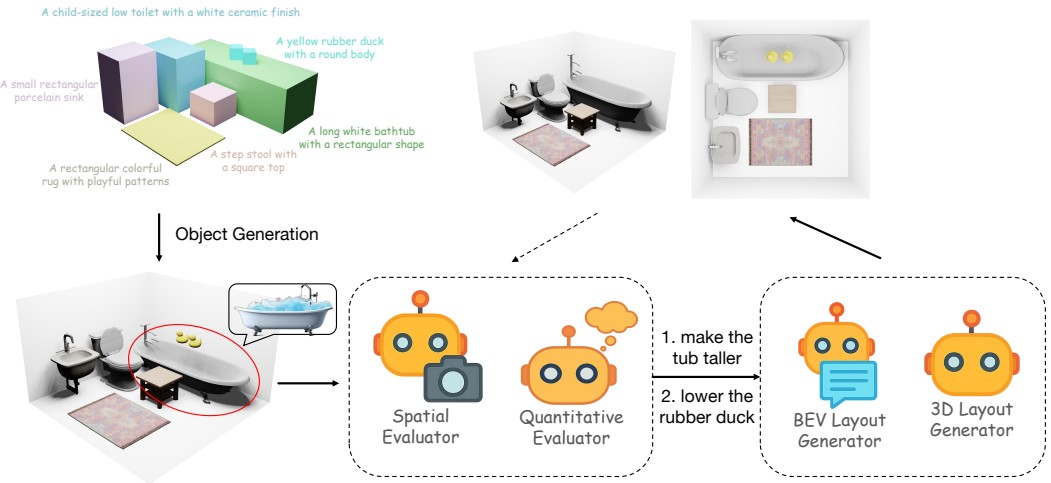

Figure 6: Illustration of Iterative Asset-Layout Alignment

# D  Inference Time

We report inference time under consistent hardware and network conditions, measured from text input to 3D layout generation. Table 4 summarizes results over 30 samples.

Table 4: Inference time comparison across methods (30 samples).

| Method | Average (s) | Min (s) | Max (s) |
|---|---|---|---|
| LayoutGPT | 5.87 | 5.32 | 6.67 |
| IDesign | 166.86 | 148.12 | 229.74 |
| Holodeck | 210.23 | 172.59 | 291.88 |
| Ours | 185.24 | 109.45 | 298.13 |

Our approach is competitive with existing baselines. A large portion of the runtime ($\sim$26 seconds per object) comes from the Trellis object generation module, which is modular and can be replaced with object retrieval. As object generation models continue to improve, this component is expected to become significantly faster.

# E  Impact Statement

This paper focuses on the technical advancements in realistic 3D indoor scene synthesis. The work aims to enhance applications in virtual reality, gaming, and design, which could have positive societal implications in these domains. However, this study does not directly address potential societal impacts, including possible negative consequences such as malicious or unintended uses (e.g., generating fake scenes), fairness considerations, privacy concerns, or security risks that might arise from the application of this technology. The paper primarily presents technical research and does not discuss the deployment of the technology or potential mitigation strategies for negative impacts.

