# OpenReview forum: "Direct Numerical Layout Generation for 3D Indoor Scene Synthesis via Spatial Reasoning"
_NeurIPS.cc/2025/Conference — NeurIPS 2025 poster_

### Official Review · Reviewer_okhS · 2025-06-11

**Clarity:** 4
**Significance:** 3
**Originality:** 3
**Rating:** 5
**Confidence:** 3

**Summary:**

This paper proposed a model that can generate renderable indoor scenes from text prompts. This is achieved by training several specialized LLM agents. The authors decompose the task into BEV layout generation, 3D lifting, and refinement, each stage handled with a specialized agent. The agents are trained with guidance from other VLMs/LLMs such as GPT4o through SFT and DPO. The authors show visual examples of the generated scenes and do and numerical comparisons with a few baselines. I find the overall quality of the generated scene pretty good and plausible. Overall, this is a technically sound and well-written paper solving a useful task, but I have some questions regarding evaluations and limitations.

**Questions:**

- How is the shape and size of the room determined? Is it a fixed size across training and validation, or input to the layout generation framework? In the real world, rooms are often not square-shaped and may have additional walls inside them. Is your model capable of handling these additional constraints?
- The scenes shown are kind of toyish and empty, with only a few objects. How well can your model scale up to more complex scenes? For example, an auditorium with hundreds of chairs, or some special types of rooms such as a conservatory in a botanical garden filled with plants? The limitations section touches on this a little bit, but I think the discussion is not enough. What is the exact boundary of complexity that the current model can push? Can we scale it up in the future with more data/more GPUs, or are there any fundamental issues you would foresee?
- I'm very interested in seeing how well the proposed method can handle the composition of objects (for example, many small objects placed on a shelf/table). In Fig 3 last row, are the objects on the table generated separately and placed on the table, or is the "table with object" a single asset? The bathtub with ducks example is interesting, but in general, I would like to see more examples of how these compositions are achieved and gradually modified to be physically plausible.

**Ethical Concerns:**

["NO or VERY MINOR ethics concerns only"]

**Final Justification:**

My concerns on evaluations are largely addressed. Although the paper still has some clear limitations, as I pointed out in the discussions, I think the paper provides valuable knowledge that outweighs the limitations.
I noticed other reviewers challenge the novelty and soundness in the LLM fine-tuning part. I'm not an expert on that so those might be valid concerns. Therefore, I list my confidence (3) to be relatively low, but I would recommend acceptance to the best of my own knowledge.

**Limitations:**

The authors discussed limitations in inference time and complexity constraints, but I want to see more in-depth discussions on the complexity constraints (see Questions). Also I would like to see some failure cases.

**Quality:**

3

**Strengths And Weaknesses:**

Strength:
- Decomposing the task into layout generation, 3D lifting, and refinement makes sense intuitively.
- Training specialized agents with SFT followed by RL-based finetuning, although being a standard pipeline for LLM finetuning, makes sense in this task.
- The paper is well-written and easy to follow.

Weaknesses:
- Evaluation: The authors do automatic evaluations with VLM in Tab 1, which is valuable, but the authors should also add human preference evaluations, which is more straightforward and convincing.
- Baselines: The authors compare against a few baselines using LLM agents, but it would also be interesting to see how rule-based methods work. For example, the method used in Infinigen-Indoors can handle pretty complex layout planning. Since there aren't many objects in the rooms generated, my guess is that the rule-based method would be a pretty strong baseline.
- There are some other limitations I find that I want to discuss, but I put them in the Questions section.

---

> ### Author Rebuttal · Authors · 2025-07-31
>
> Thank you sincerely for your valuable time and insightful comments.
>
> ### **W1: Human preference evaluation**
>
> Thank you for the suggestion. In response, we conduct a **user study with 25 compensated volunteers**, all with relevant backgrounds in 3D vision or computer science.  Participants are asked to rate anonymized, randomly ordered layouts generated by different methods across the following three criteria:
>
> Semantic Compliance: How well the layout matches the input text description.
>
> Layout Rationality: Whether the spatial arrangement of objects is reasonable, functional, and typical of real-world scenes.
>
> Physical Plausibility: Whether the layout is physically sound—for example, no floating or interpenetrating objects, and appropriate support relationships.
>
> | Method              |   Semantic Compliance |   Physical Plausibility |   Layout Rationality |
> |:--------------------|----------------------:|------------------------:|---------------------:|
> | LayoutGPT (coarse)  |                  3.72 |                    1.84 |                 1.72 |
> | LayoutGPT (medium)  |                  2.88 |                    1.64 |                 1.76 |
> | LayoutGPT (fine)    |                  3    |                    2.32 |                 2.04 |
> | LayoutGPT (overall) |                  3.2  |                    1.93 |                 1.84 |
> | I-Design (coarse)   |                  3.72 |                    2.72 |                 2.44 |
> | I-Design (medium)   |                  3.08 |                    3.36 |                 2.48 |
> | I-Design (fine)     |                  3.32 |                    3.72 |                 2.72 |
> | I-Design (overall)  |                  3.37 |                    3.27 |                 2.55 |
> | Holodeck (coarse)   |                  4.12 |                    3.64 |                 2.84 |
> | Holodeck (medium)   |                  2.8  |                    2.52 |                 2.48 |
> | Holodeck (fine)     |                  2.4  |                    2.6  |                 2.04 |
> | Holodeck (overall)  |                  3.11 |                    2.92 |                 2.45 |
> | Ours (coarse)       |                  4.56 |                    4.64 |                 4.52 |
> | Ours (medium)       |                  4.6  |                    4.68 |                 4.64 |
> | Ours (fine)         |                  4.84 |                    4.72 |                 4.8  |
> | Ours (overall)      |                  4.67 |                    4.68 |                 4.65 |
>
> Our method consistently received the highest average scores across all categories, showing strong agreement with human preferences. This study complements the VLM-based evaluation and confirms that our method produces layouts that are not only quantitatively strong but also preferred by human evaluators.
>
> We also include comparison with ATISS, DiffuScene, and Infinigen:
>
> | Method     |   Physical Plausibility |   Layout Rationality |
> |:-----------|------------------------:|---------------------:|
> | ATISS      |                    3.66 |                 3.54 |
> | Infinigen  |                    4.66 |                 4.44 |
> | Diffuscene |                    3.46 |                 3.1  |
> | Ours       |                    4.62 |                 4.48 |
>
> While baselines perform strongly overall, our method matches or surpasses their performance on certain scene categories, and additionally supports open-vocabulary generation.
>
> ### **W2: Rule-based baseline comparison (e.g., Infinigen-Indoors)**
>
> We include **Infinigen-Indoors**, ATISS, and DiffuScene as new baselines and compared its performance on bedrooms and living rooms with our method:
>
> | Method     | Scene      | Pos   | Rot   | PSA  |
> |------------|------------|-------|-------|------|
> | ATISS      | bedroom    | 81    | 70    | 86   |
> | ATISS      | livingroom | 71.5  | 79.5  | 76   |
> | ATISS      | overall    | 76.25 | 74.75 | 81   |
> | Diffuscene | bedroom    | 66    | 63    | 70   |
> | Diffuscene | livingroom | 66.5  | 66    | 60   |
> | Diffuscene | overall    | 66.25 | 64.5  | 65   |
> | Infinigen  | bedroom    | 75.5  | 78.3  | 80   |
> | Infinigen  | livingroom | 75.5  | 75.8  | 76   |
> | Infinigen  | overall    | 75.5  | 77.05 | 78   |
> | Ours       | bedroom    | 82.5  | 75    | 96   |
> | Ours       | livingroom | 86    | 75.4  | 80   |
> | Ours       | overall    | 84.25 | 75.2  | 88   |
>
> We also evaluate Infinigen in the **user study**. While it performs well, our method achieves similar or better results on specific scene types and supports **open-vocabulary generation**, which rule-based systems cannot easily accommodate.
>
> ### **Q1: Room shape and input**
>
> The room size is provided as input to the layout generator. In CoT Activation, we extract it from 3D-Front scene bounding boxes as part of the input in training. In DPO, the object size is inferred from the LLM during data pair generation, as shown in Appendix A.1. Specifically, the LLM predicts the appropriate size based on the scene description, and this inferred size is then used together with the scene context to construct data pairs for DPO training. Currently, we support rectangular rooms, which cover the majority of real-world layouts. Non-rectangular or complex-shaped rooms are an interesting future extension, though they are not the main focus of our method.
>
> ### **Q2: Handling complex scenes**
>
> In practice, our model performs well in scenes with up to 30 objects, which shows no significant difference compared to the baselines. This upper limit stems from the training data distribution (3D-Front rarely contains denser scenes), rather than algorithmic constraints. We believe the method is scalable, particularly through training on datasets where individual scenes contain more objects, as well as by incorporating strategies such as recursive layout decomposition, which divides complex rooms into manageable subregions. We are actively exploring these directions. We will add more discussions in the revised paper.
>
> ### **Q3: Object composition and substructures**
>
> In Figure 3 (last row), each item on the table is placed individually. Due to rebuttal formatting limits, we could not include more visualizations, but will provide extensive examples of such compositions in the next version to offer a more intuitive understanding of Iterative Asset-Layout Alignment process.
>
> We hope our responses can address your concerns. Please feel free to inform us of any further issues.

---

> > ### Comment · Reviewer_okhS · 2025-08-01
> > **discussion**
> >
> > Thank the authors for the rebuttal.
> >
> > I find the additional human studies convincing. Although a few numbers are worse than Infinigen, the method overall outperforms the baselines, also confirmed by the visualizations shown in the initial submission.
> >
> > The system only supports rectangular rooms and a limited number of objects. These are obvious limitations that limit the practical usability of the system. That being said, I do agree with the authors that these can be solved by adding more data. It is reasonable to list these as future works.

---

### Official Review · Reviewer_QFty · 2025-07-02

**Clarity:** 3
**Significance:** 2
**Originality:** 2
**Rating:** 5
**Confidence:** 4

**Summary:**

The paper proposes DirectLayout, a framework that generates 3D indoor layouts directly from textual descriptions. Unlike existing methods that either regress numerical layouts from data (with poor generalization) or use intermediate symbolic representations (with rigidity and poor fine-grained control), DirectLayout introduces a three-stage pipeline:

1. BEV layout generation, enhanced with Chain-of-Thought (CoT) reasoning.
2. 3D lifting using commonsense priors.
3. Iterative asset-layout alignment using feedback from dual evaluators (LLM + VLM).

**Questions:**

How sensitive is the performance of your BEV Layout Generator to the quality of the GPT-4o-generated CoT traces? Did you experiment with noisy or incorrect CoT traces to evaluate robustness?
Your CoT-Grounded Layout Reward combines both VLM and LLM feedback. How critical is this dual-evaluator setup?
How well does DirectLayout generalize to unusual room types or unseen object categories not represented in the 3D-Front dataset or training prompts?
What are the most common failure modes you’ve observed?

**Ethical Concerns:**

["NO or VERY MINOR ethics concerns only"]

**Final Justification:**

All my concerns have been addressed. I appreciate the authors' efforts. I would still encourage the author to discuss the limitations of BEV representations. I am leaning towards accepting this paper.

**Limitations:**

Yes. But there are more limitations, especially on the usage of BEV as the intermediate representation.

**Quality:**

2

**Strengths And Weaknesses:**

The paper clearly defines the limitations of existing paradigms and motivates its simple yet effective algorithm based on Chain-of-Thought (CoT) reasoning and numeric generation. The model outperforms LayoutGPT, Holodeck, and I-Design across nearly all metrics.

The main weakness of the paper is the lack of evaluation on out-of-distribution generalization to novel categories of scenes not present in the training set. In particular, the method relies on heavy fine-tuning through Supervised Fine-Tuning (SFT) and Direct Preference Optimization (DPO), which may hinder its generalization compared to methods that keep the base LLM frozen.

Another limitation is that, being based on a Bird’s-Eye View (BEV) representation, the method is not well-suited for handling scene layouts with significant vertical stacking or clutter—such as books on bookshelves. While this is not the primary focus of the paper, it does limit its broader applicability.

Finally, although the authors use LLM-based evaluators (as in LayoutVLM), they do not report any human studies to directly assess alignment or plausibility.

---

> ### Author Rebuttal · Authors · 2025-07-31
>
> Thank you sincerely for your valuable time and insightful comments.
>
> ### **W1 & Q3: Lack of out-of-distribution (OOD) evaluation; impact of SFT/DPO on generalization**
>
> We appreciate the reviewer’s insightful concern. In fact, the qualitative and quantitative evaluations in both the paper and appendix already assess generalization to unseen scene types. During the Supervised Fine-Tuning (SFT) stage, we use data derived from 3D-Front, which only contains four scene categories: bedroom, living room, study, and dining room. However, the scene prompts used in DPO training and final evaluations are generated separately and explicitly exclude those scene types, ensuring no category-level overlap between training and evaluation. This separation was enforced via two disjoint prompt sets detailed in Appendix A.3.
>
> ### **W2: BEV limitation in handling vertical stacking (e.g., books on bookshelves)**
>
> We agree with the reviewer that BEV has inherent limitations in modeling vertical stacking. However, many such cases (e.g., “books on bookshelf”) can be treated as composite assets in our system. For example, “a filled bookshelf” can be generated as a single object, which captures the visual semantics without modeling each individual book.
>
> Moreover, in our experiments, we find that explicitly separating BEV layout and 3D lifting leads to better overall performance than jointly modeling them, highlighting the effectiveness of our two-stage design. In practice, scenes with extremely dense vertical stacking (e.g., cluttered desktops or fully packed shelves) are relatively rare and not the primary focus of our work. That said, fully modeling fine-grained vertical clutter remains an important future direction, which may require more powerful spatial reasoning capabilities.
>
> ### **W3: No human study reported to directly assess alignment or plausibility**
>
> We thank the reviewer for pointing this out. In addition to automated evaluations (e.g., via LayoutVLM), we conduct a human evaluation study to directly assess alignment and plausibility from a user-centric perspective. Specifically, we recruite 25 compensated participants with relevant backgrounds in computer science and 3D vision. Each participant evaluated layout samples from multiple methods, randomized and anonymized, based on three axes (All ratings are collected using a 5-point Likert scale):
>
> Semantic Compliance: Whether the layout reflects the input prompt faithfully.
>
> Physical Plausibility: Whether objects are well-supported and physically reasonable.
>
> Layout Rationality: Whether the spatial arrangement is typical and convincing.
>
> | Method              |   Semantic Compliance |   Physical Plausibility |   Layout Rationality |
> |:--------------------|----------------------:|------------------------:|---------------------:|
> | LayoutGPT (coarse)  |                  3.72 |                    1.84 |                 1.72 |
> | LayoutGPT (medium)  |                  2.88 |                    1.64 |                 1.76 |
> | LayoutGPT (fine)    |                  3    |                    2.32 |                 2.04 |
> | LayoutGPT (overall) |                  3.2  |                    1.93 |                 1.84 |
> | I-Design (coarse)   |                  3.72 |                    2.72 |                 2.44 |
> | I-Design (medium)   |                  3.08 |                    3.36 |                 2.48 |
> | I-Design (fine)     |                  3.32 |                    3.72 |                 2.72 |
> | I-Design (overall)  |                  3.37 |                    3.27 |                 2.55 |
> | Holodeck (coarse)   |                  4.12 |                    3.64 |                 2.84 |
> | Holodeck (medium)   |                  2.8  |                    2.52 |                 2.48 |
> | Holodeck (fine)     |                  2.4  |                    2.6  |                 2.04 |
> | Holodeck (overall)  |                  3.11 |                    2.92 |                 2.45 |
> | Ours (coarse)       |                  4.56 |                    4.64 |                 4.52 |
> | Ours (medium)       |                  4.6  |                    4.68 |                 4.64 |
> | Ours (fine)         |                  4.84 |                    4.72 |                 4.8  |
> | Ours (overall)      |                  4.67 |                    4.68 |                 4.65 |
>
> Our method consistently received the highest average scores across all categories, showing strong agreement with human preferences. This study complements the VLM-based evaluation and confirms that our method produces layouts that are not only quantitatively strong but also preferred by human evaluators.
>
> We also include comparison with ATISS, DiffuScene, and Infinigen:
>
> | Method     |   Physical Plausibility |   Layout Rationality |
> |:-----------|------------------------:|---------------------:|
> | ATISS      |                    3.66 |                 3.54 |
> | Infinigen  |                    4.66 |                 4.44 |
> | Diffuscene |                    3.46 |                 3.1  |
> | Ours       |                    4.62 |                 4.48 |
>
> While baselines perform strongly overall, our method matches or surpasses their performance on certain scene categories, and additionally supports open-vocabulary generation.
>
> ### **Q1: Robustness to noisy CoT traces**
>
> We conduct controlled experiments to assess robustness by:
>
> * **Shuffling** the order of CoT steps, and
> * **Removing** the CoT reasoning entirely.
>
> | Setting      | Pos   | Rot   | PSA   | OOB↓  | Collision↓ | Clip  |
> | ------------ | ----- | ----- | ----- | ----- | ---------- | ----- |
> | Normal CoT   | 64.47 | 57.18 | 68.82 | 26.04 | 16.77      | 25.27 |
> | Shuffled CoT | 60.33 | 54.20 | 62.90 | 29.40 | 16.90      | 22.14 |
> | No CoT       | 56.80 | 50.95 | 54.75 | 34.60 | 18.55      | 21.76 |
>
> Shuffling degrades performance modestly, while removing CoT causes a significant drop. These results show that structured CoT reasoning is essential for layout quality.
>
> ### **Q2: Importance of dual reward evaluators**
>
> Our reward signal integrates:
>
> * VLM-based evaluator focuses on high-level visual and physical plausibility, judging the overall realism and global coherence of the generated layout.
>
> * Reasoning LLM-based evaluator operates at a fine-grained, object-level, verifying that each object's position, orientation, and relation to others.
>
> These components are complementary. Our ablations show that removing either leads to imbalanced layout optimization. As shown in Table 2, the full model using a dual-evaluator significantly outperforms the simple reward setting that relies on a single evaluator. Thus, the dual-evaluator setup is critical for robust and generalizable learning.
>
>
> We hope our responses can address your concerns. Please feel free to inform us of any further issues.

---

> > ### Comment · Reviewer_QFty · 2025-08-05
> >
> > All my concerns have been addressed. I appreciate the authors' efforts. I would still encourage the author to discuss the limitations of BEV representations.

---

> > > ### Author Response · Authors · 2025-08-06
> > >
> > > Thank you for the suggestion. We will discuss the limitations of BEV representations in a revision.

---

### Official Review · Reviewer_A1tB · 2025-07-02

**Clarity:** 1
**Significance:** 3
**Originality:** 2
**Rating:** 3
**Confidence:** 4

**Summary:**

This paper studies 3D layout generation. The proposed pipeline follows the widely used LLM-agent pipeline, trained SFT using GPT-generated CoT data, and then standard DPO. The contributions of this paper mainly lie in the engineering tricks within this pipeline, including the specific reward designed for the alignment between CoT and the generated layout and the final interactive asset-layout alignment for refinement. Both qualitative and quantitative measurements show that the proposed method is effective.

**Questions:**

- I wonder if GPT-4o can generate reasonable CoTs, instead of just generating perfunctory analysis to overfit the answer. It would be beneficial to provide examples of GPT-generated CoTs.
- The proposed method leverages the visual grounding of a trained model to generate a layout, which reasons on top of the coordinates. I would like to see the raw outputs of the planner, to see how it reasons with the coordinates, and how precise they are.
- Please provide more explanations about the seven terms in "Object-Level Criteria".
- If the author(s) disagree with the reviewer's claim that "the novelty of the paper is engineering tricks on top of a widely adopted framework", please argue the novelties that can be regarded as the novelty compared with the whole LLM agent field to improve general LLM agent designs, instead of the novelties in the layout generation task.

**Ethical Concerns:**

["NO or VERY MINOR ethics concerns only"]

**Final Justification:**

I would like to maintain the current score, as the current paper has significant issues in both writing and reliability of LLM/VLM-based evaluation, and concerns about model capability.

**Limitations:**

The authors mentioned the limitations. Some others mentioned in "Weaknesses" about the lack of 3D z-axis capability.

**Paper Formatting Concerns:**

N/A.

**Quality:**

3

**Strengths And Weaknesses:**

### Strengths
- The high-level ideas of the method are straightforward to understand. The engineering tricks are relatively well-motivated.
- All the visualizations in this paper are impressive, showing that the generated layout is reasonable (while also thanks to the out-of-the-box object generator).
- Both quantitative and qualitative scores show that the proposed method is effective and outperforms state-of-the-art.
- Specifically, the experiment part considers the use of different LLMs and tests the proposed method in different settings, e.g., open-source only, to make a fair and holistic comparison.

### Weaknesses
- As the paper employs the widely adopted LLM agent pipeline, with CoT as also a widely adopted extension, even if it might be the first to employ it in layout generation, transferring such a well-studied pipeline to the new task should not be solely regarded as a contribution/novelty.
- **Crucial.** The method section is not well-written. Based on such a widely used framework, the actual novelties of this paper are engineering tricks. This makes the paper more like a technical report, revealing numerous details; however, on the other hand, these details are still mainly text descriptions with uninformative formulas, which are still far from being reproduced.
    - For example, Sec. 3.1.2's "Object-Level Criteria" introduces the reward function with seven terms. This is very detailed, but okay, as they are related to the main contribution.
    - However, none of these terms are defined rigidly or explained in detail - simply mentioning "Measures the reasonableness of spatial relations with related objects" or "Checks that spacing between objects supports accessibility and func162 tionality. Overlaps may be acceptable if vertical stacking is plausible" is far from sufficient, which cannot tell how they are defined and distinguished.
    - All the formulas (3)-(7) are just uninformative and simple rephrases of texts. For example, (3) is just a simple yet space-consuming explanation of "the proportion of objects in the layout that satisfy the corresponding requirement" (L169). All those formulas are just placeholders to satisfy "there should be a formula".
    - All these make the paper hard to reproduce, even if the engineering tricks are the focus of the paper.
- Like all LLM agent methods, the inference requires a full round between the LLM planner and the layout generator, which can be time-consuming. Specifically, the proposed method involves an object generator for per-object generation, which further increases the time consumption.
- As using BEV, the proposed method may not be able to generate complicated z-axis, e.g., a large wardrobe hanging above a table with several fruits on it, under which there is a chair. As shown in the supplementary video "bathroom_wo_refine", the model seems to be confused about the height location of the object. This hurts the scope of the proposed method, raising concerns of whether the proposed method is still only generating 2D/pseudo-3D layouts instead of real 3D.

---

> ### Author Rebuttal · Authors · 2025-07-31
>
> Thank you sincerely for your valuable time and insightful comments.
>
> ### **W1 & Q4: Novelty of using LLM agent pipeline and CoT**
>
> Our primary contribution and key insight lie in presenting a new perspective specifically for the 3D scene synthesis domain. Prior approaches (e.g., ATISS) mainly rely on fully supervised training over limited datasets, which learn fixed spatial distributions and thus struggle to generalize to new object categories or diverse scenes. Recent efforts (e.g., Holodeck, I-Design) have begun leveraging large language models (LLMs) to address this gap, but these methods often depend on manually defined spatial constraints and external solvers, which limit their flexibility and ability to handle fine-grained spatial relations. It is widely recognized that LLMs find it challenging to directly generate precise numerical layouts. In contrast, our approach integrates CoT reasoning with DPO to significantly enhance the LLM’s generalizable spatial reasoning capabilities, enabling it to produce complete numerical 3D layouts directly. This opens new avenues for thinking about generative 3D scene layout design.
>
> Moreover, for LLM agent, we use an entropy-weighted reward integration mechanism that dynamically balances multiple layout quality criteria. While most existing LLM research focuses on domains like mathematics or programming, 3D spatial reasoning is an important and challenging application scenario. Our work demonstrates a practical approach for enabling LLMs to tackle generative 3D spatial reasoning tasks. To facilitate further research, we plan to release the CoT dataset and source code upon paper acceptance.
>
> ### **W2 & Q3: Method section clarity and reproducibility**
>
> We acknowledge that the method section can be improved for clarity. We sincerely appreciate the valuable feedback and will incorporate these improvements in the subsequent version.
>
>
> (1) **On evaluation criteria definitions**:
>   The spatial and functional “criteria” mentioned (e.g., “reasonableness of spatial relations”, “accessibility and functionality”) serve as guiding directions for evaluators rather than rigidly   defined quantitative metrics. These criteria are clarified with concrete prompt examples in Appendix A.1. We also provide more detailed explanations:
>
> Relative Alignment (C1)
> This criterion assesses the geometric congruence and spatial relationship between related objects. It evaluates whether objects that are functionally or conceptually linked exhibit appropriate alignment, such as central, horizontal, or vertical congruity, or organized arrangements like grids, based on their intended spatial interaction.
>
> Global Positioning (C2)
> This criterion evaluates the semantic appropriateness of an object's placement within the overarching room context. It assesses whether the object's position aligns with conventional spatial norms and functional requirements relative to the entire layout, ensuring it contributes to a coherent and functional environment.
>
> Consistency with CoT (C3)
> This criterion verifies the fidelity of object placements against any explicit or inferred spatial instructions provided within a Chain-of-Thought (CoT) description. It ensures that the physical arrangement of objects in the scene directly corresponds to the textual guidance, identifying discrepancies where the visual representation deviates from the described intent.
>
> Inter-object Distance (C4)
> This criterion measures the functional spacing between distinct objects, ensuring adequate separation for accessibility, navigability, and intended use. While minimal bounding box overlaps may be permissible in instances of plausible vertical stacking (e.g., an object placed on another), the primary assessment focuses on maintaining sufficient clearance for unimpeded interaction and movement.
>
> To manage space in this rebuttal, we provide detailed explanations for four key criteria (C1–C4). If you’d like, we’d be happy to provide the full descriptions of all seven criteria in a follow-up response or the revised version.
>
> (2) **On formulas (3)-(7)**:
>   These formulas are intended to clearly explain the detailed process of reward computation, aiming to enhance the clarity and reproducibility of the paper. However, we acknowledge that some formulas overlap with the textual descriptions. For example, Equation (3) can be more concisely described in words, and Equation (5) corresponds to entropy calculation, which can also be expressed textually. On the other hand, Equation (4) is difficult to describe clearly using only words, while Equations (6) and (7) could be combined but represent a complex weighting calculation that cannot be fully captured in simple language. We will streamline the paper by removing redundant overlaps between text and formulas, thereby improving the clarity and conciseness of the presentation.
>
> (3) On reproducibility:
>   All prompts and detailed experimental parameters are provided in the appendix. We will open source code and data upon acceptance to ensure full reproducibility.
>
> ### **W3: Inference time concerns**
>
> Our inference time, measured under consistent hardware and network conditions, covers the full pipeline from text input to 3D layout generation. As shown in the table (30 samples), our method performs competitively with baselines (Unit: seconds). Object generation (Trellis) takes ~26 seconds per object, but it's modular and can be replaced with object retrieval. As generation tools improve, this time is expected to drop. We don't consider this to be a fundamental bottleneck of our method.
>
> | Method                      |   Average |   Min |   Max |
> |:----------------------------|---------------------------:|-----------------------:|-----------------------:|
> | LayoutGPT                   |                       5.87 |                   5.32 |                   6.67 |
> | IDesign (~60% failure rate) |                     166.86 |                 148.12 |                 229.74 |
> | Holodeck                    |                     210.23 |                 172.59 |                 291.88 |
> | Ours                        |                     185.24 |                 109.45 |                 298.13 |
>
>
> ### **W4: Handling of z-axis and true 3D layout**
>
> While BEV layouts represent objects as bounding boxes in the XY-plane, our framework allows overlapping bounding boxes to reflect vertical stacking. We leverage LLMs’ strong priors on object height ranges and relationships, enabling accurate height estimation despite the 2D base layout. As shown in Table 2, decomposing the problem (BEV layout + height reasoning) outperforms direct 3D generation attempts.
>
> The issue in the bathroom_wo_refine example is not a general failure in height prediction, but a rare case of incorrect object perception, the model initially assumed the bathtub was filled with water, leading it to place the rubber duck on top. To address such cases, we introduce an Iterative Asset-Layout Alignment module that refines the layout based on feedback from generated assets, successfully correcting the error in later iterations. This is the result of the ablation experiment (not our full model) and our framework is specifically designed to handle this. Therefore, this case should not be considered a main weakness of our method, nor does the ablation visualization reflect a fundamental flaw.
>
>
> ### **Q1: CoT Reasoning Quality**
> To demonstrate the quality of reasoning, we provide below an example of a GPT-4o–generated CoT  (only a partial example shown due to space constraints):
> ```
> "Entity Extraction": "The scene contains: 2 multi-seat sofas, 1 armchair, 1 coffee table, 2 console tables, 1 cabinet, and 2 stools. These pieces suggest a living room setup focused on seating and moderate storage/display furniture.",
>
> "Order Decision": "First, we place the two multi-seat sofas since they are the largest and central to the scene. Next, we position the coffee table in relation to the sofas to form a functional conversation area...",
>
> "Spatial Reasoning": "The layout starts by defining the primary seating area. A multi-seat sofa is placed along the right side of the room, oriented vertically to establish one edge of the seating space. Its center is set around (162px, 176px), with dimensions of 60px in length and 25px in width, rotated -90 degrees. To complement this, a second multi-seat sofa is positioned horizontally below, forming an L-shaped arrangement. It is centered at (110px, 133px), maintaining a 0-degree orientation, creating a natural corner ideal for conversation..."
> ```
> This example shows that GPT-4o produces structured and context-aware reasoning. The CoT is not superficial, it breaks down the task and considers spatial relationships rather than overfitting the final answer.
>
> ### **Q2: Raw Planner Output**
> Below is an example of raw planner output, showcasing its multi-step reasoning process (only a partial example shown due to space constraints):
> ```
> "Entity Extraction": "The objects that should be placed in the scene include a multi-seat sofa, a lounge chair, a coffee table, a TV stand, a dining table, multiple dining chairs, and pendant lamps."
>
> "Order Decision": "The order of placement should start with the largest and most central objects, which are the multi-seat sofa and the dining table, as they likely serve as the focal points of the living and dining areas..."
>
> "Spatial Reasoning": "To begin shaping the room, a multi-seat sofa is positioned as the anchor of the seating area. It’s placed along the right side of the space, rotated vertically to make efficient use of the wall without cutting into circulation paths. With a generous width and length—measuring 79px by 34px—it defines the main visual boundary of the living zone. The center is set at (194px, 65px), maintaining openness in the central floor area..."
> ```
>
> We hope our responses can address your concerns. Please feel free to inform us of any further issues.

---

> ### Comment · Reviewer_A1tB · 2025-08-06
>
> I sincerely appreciate the authors for their rebuttal.
>
> - The concerns on LLM/CoT is partially addressed. This can be regarded as an (engineering) contribution to make it work, but novelty is still limited.
> - About the clarity of the method part, the original text does not show that they are "guiding directions for (human) evaluators" - which can be guided by a "coarse direction", but are for automated evaluators - which do require a rigid definition. Also, the computation of the evaluation involves per-bject checking, which seems difficult to be performed by human. Even now, I am not sure how those evaluators are implemented.
> - The concerns on the inference time are addressed.
> - About the artifact in "bathroom_wo_refine", I am not evaluating it as it is from the full model, but considering the actual reason behind "why this requires refinement to prevent failure" and "whether this refinement is just a workaround for the fundamental limitation of lacking z-axis capability." I am also not convinced by the explanation of "the model initially assumed the bathtub was filled with water, leading it to place the rubber duck on top," because the duck is much higher than the bowl of the bathtub. I would like some further clarifications on how frequent this pattern and refiner works.
> -  The raw outputs of the LLM addresses my concerns on this. The GPT-4o's CoT is just what I expected, tries to generate a explanation that fits the answer; however, when a human actually plans, they may have a top-down or coarse-to-fine approach, or some revisions of exisiting plans when finding a later object cannot fit. This GPT CoT is just like a lucky person who magically puts all objects on all correct places without planning or retrying. I hope the actual model's CoT could be more powerful after RL training.

---

> ### Author Response · Authors · 2025-08-06
>
> Thank you once again for your thoughtful feedback and for taking the time to review our rebuttal. We would like to address the remaining concerns in your latest response.
>
> **1. On Novelty: Beyond Engineering Contributions**
> We would like to re-emphasize that our contribution lies in proposing a new perspective for 3D indoor scene synthesis—a novel solution distinct from previous optimization-based methods (as explained in the previous rebuttal), and is not merely an engineering effort. Our work primarily addresses challenges in 3D indoor scene synthesis, not LLM agents. Therefore, more attention should be given to its contributions to the field of 3D indoor scene synthesis.
>
> **2. On Evaluation Criteria**
> We appreciate your concern about the clarity and definition of the evaluation criteria. Our evaluators are implemented via reasoning LLM and VLM, not human, but they are instructed to mimic human judgment using semantically grounded criteria. While many tasks require rigid evaluation metrics, layout generation is inherently ambiguous and subjective. In our experience, using these criteria allows the evaluator model to reason in a human-like manner and yield consistent and practical judgments. For instance, in the case of inter-object distance: when a bed and nightstand heavily overlap in BEV, the evaluator uses physical priors to infer whether the overlap is likely due to vertical stacking. In this case, the evaluator would consider it unreasonable, as both objects are typically placed directly on the floor. We believe that overly rigid definitions may misclassify novel yet valid layouts. It is also difficult to rigidly define what constitutes a reasonable object placement, as it is nearly impossible to summarise or exhaustively enumerate all types of object interactions, spatial relationships, and functional requirements. That said, we agree that clearer documentation would help. **As stated in our rebuttal, we have provide prompt examples in Appendix A.1 (this is exactly what we use in our practice, and it intuitively demonstrates how those evaluators are implemented). If you would find it helpful, we are happy to further provide: raw outputs from evaluators, or an example BEV layout to try it yourself, so you may directly assess how these judgments are made.**
>
> **3. On the "Bathroom" Example and Role of Refinement**
> Thank you for your observation on the bathroom_wo_refine case. We now provide a clearer explanation of the visual artifact:
> As shown in the supplementary figure, the bathtub asset includes a vertical shower fixture, which is considered part of the bathtub model. In the numeric layout, the rubber duck is placed with a z-range of 0.45–0.55, on top of the bathtub’s bounding box (z = 0–0.5). Since the visualizer renders the shower head and the duck at similar height ranges, the duck appears disproportionately high—even higher than nearby bowl. This is a side effect of not accounting for asset-specific geometry during initial layout generation.
> **This issue arises not from a fundamental limitation in z-axis modeling, but from the inability of the layout planner to directly perceive asset geometry.** Our refinement module addresses precisely this challenge, bridging the gap between symbolic layout reasoning and asset-level appearance grounding.
> To quantify the frequency of such issues and the effectiveness of refinement, we ran 20 test scenes:
> * 13/20 scenes required no refinement
> * 5/20 were corrected with 1 round of refinement
> * 2/20 required 2 refinement rounds
>
> These results indicate that while most scenes are resolved without refinement, the refinement module plays a crucial role in improving alignment when necessary. We will incorporate these findings more clearly in a revision.
>
> Please let us know if you have any further questions.

---

> > ### Comment · Reviewer_A1tB · 2025-08-08
> >
> > I appreciate the authors for their follow-ups.
> >
> > As for the evaluation part, I think this implementation is more like a LLM/VLM-based VQAScore, which may not be able to fully assess some very low-level or abstract properties. For example, it is hard even for human to check every object's orientation or size, so LLM or VLM may also not check every object carefully but mainly the large objects. Specifically, the Quantitative Evaluator may have different behaviors or rubrics under different random seeds of the LLM/VLM. Still, I would suggest the authors to significantly revise this part to indicate how the evaulator LLM/VLM works.
> >
> > As for the `bathroom_wo_refine`, I think the author mentioned a limitation of their method: "inability of the layout planner to directly perceive asset geometry" and some (2/20) objects even requires multiple rounds of refinements. I appreciate the authors' honesty, but I would like to know if there are any possible ways to mitigate this, e.g., let the planner see the image of the objects.
> >
> > At this point, I would like to maintain the current score. I still welcome more discussions on these points.

---

> > > ### Author Response · Authors · 2025-08-09
> > >
> > > We thank you for your feedback.
> > >
> > > * We use reasoning LLM to check low-level attributes. Due to their rich common-sense knowledge, strong mathematical abilities, and allowance for long chains of thought, they can examine object orientation, sizes, and more. Moreover, we use a unified random seed for evaluating all samples. Comparative experimental results and user study (already presented in our rebuttal to other reviewers) demonstrate the effectiveness of our approach, and ablation studies further validate the effectiveness of our reward. Below is an raw example of the reasoning chain from the LLM to help illustrate the process of checking each object’s size:
> > > ```
> > > Size proportion:
> > >
> > > Sofa (79×34) vs Coffee table (49×49):
> > > The coffee table is smaller in both dimensions than the sofa. This matches real-world expectations, where a sofa is a much larger focal object and the table sits in front of it. Yes.
> > >
> > > TV stand (70×15) vs TV (70×15):
> > > The TV and stand have identical bounding box dimensions. In reality, a stand should extend beyond the TV in at least one dimension to provide proper support and visual balance. Having them the same size means they fully overlap in BEV, which is structurally and visually unrealistic. No.
> > >
> > > Dining table (75×34) vs Dining chairs (5×20):
> > > The chairs are extremely short in one dimension (5px) compared to the table’s length (75px). Even if 20px represents the seat depth, a 5px width for the chair front-to-back footprint is too small relative to a table’s scale. This suggests a size mismatch—chairs should occupy more space in at least one dimension for comfortable seating. No.
> > >
> > > Coffee table (49×49) vs Lounge chair (24×25):
> > > The lounge chair is smaller in both width and depth compared to the coffee table, which is plausible. Lounge chairs typically occupy less floor space than a central coffee table. Yes.
> > >
> > > Pendant lamps (26×26) over Coffee table (49×49) and Dining table (75×34):
> > > In both cases, the lamp footprint is much smaller than the table area, which is correct for overhead lighting fixtures. Yes.
> > > ```
> > > From the example, it is clear that the reasoning LLM examines the sizes of all objects (including small ones) and provides reasonable feedback.
> > > For the method description, we will improve the clarity according to your suggestions.
> > >
> > > * "Inability of the layout planner to directly perceive asset geometry" is the limitation of the layout generator, this is not a limitation of our method. Refinement is an integral part of our approach and addresses this problem. The additional reasoning time introduced by refinement does not cause a significant difference in inference time compared to the baselines (as shown in inference time table). It is indeed possible that directly perceiving asset images could mitigate some of these issues, but VLMs alone struggle to fully model object interactions and physical properties purely from asset images. Moreover, most existing 3D object generation methods cannot guarantee consistency in asset orientation, which limits the feasibility of directly generating the final layout from asset perception. From our practice, detecting and correcting errors is more practical than attempting to avoid them entirely.
> > >
> > > Regarding the “crucial weakness” you mentioned, our method being hard to reproduce, we have pointed out that all prompts can be found in Appendix A.1, training parameter details can be found in Appendix A.2, and we have provided examples to demonstrate the effectiveness of our implementation. Do you have any further questions?

---

> > > > ### Comment · Reviewer_A1tB · 2025-08-09
> > > >
> > > > I appreciate the authors for their feedback.
> > > >
> > > > ---
> > > > As for the LLM evaluation, I think the most unreliable part of LLMs is not that "it does not have those capabilities", or "whether it can." Instead, "whether it actually uses these capabilities to do the desired thing" is the most unreliable part -- even if the LLM has the capability to use CoT check logical relationship and math, there is no guarantee at that it will really do this, instead of directly assuming the result at beginning based on a "first-glance" and then trying to find support for it. For example, **one example** provided in your response is:
> > > >
> > > > > Pendant lamps (26×26) over Coffee table (49×49) and Dining table (75×34):
> > > > > In both cases, the lamp footprint is much smaller than the table area, which is correct for overhead lighting fixtures. Yes.
> > > >
> > > > However, if it follows the rubric of **another example** :
> > > > > Dining table (75×34) vs Dining chairs (5×20):
> > > > > The chairs are extremely short in one dimension (5px) compared to the table’s length (75px). Even if 20px represents the seat depth, a 5px width for the chair front-to-back footprint is too small relative to a table’s scale. This suggests a size mismatch—chairs should occupy **more space in at least one dimension for comfortable seating**. No.
> > > >
> > > > It will generate the evaluation in the opposite direction:
> > > > > Pendant lamps (26×26) over Coffee table (49×49) and Dining table (75×34):
> > > > > While the lamp is smaller than both tables, its footprint is more than half the width of the coffee table, which might **feel too dominant** when hung low above the coffee table, creating a cramped visual effect. No.
> > > >
> > > > Which shows more LLM's common sense reasoning capability -- which is not utilized when generating evaluation of the first example. Fixing the random seeds only ensure the **reproducibility** but not **behavior** or **rubric**, and can still not make the LLM give consistent and fair evaluations.
> > > >
> > > > ---
> > > > > "Inability of the layout planner to directly perceive asset geometry" is the limitation of the layout generator, this is not a limitation of our method. Refinement is an integral part of our approach and addresses this problem.
> > > >
> > > > However, the title of this paper is "**Direct Numerical Layout Generation** for 3D Indoor
> > > > Scene Synthesis via Spatial Reasoning. As claimed at L56, "We propose a novel framework for 3D indoor scene synthesis that **generates layouts** directly from text descriptions, bypassing the need for constrained optimization."
> > > >
> > > > Therefore, the authors cannot claim that the layout generator as a core contribution's limitation is not the limitation of the method -- the only case making the claim reasonable is when the layout generator is an out-of-the-box application of another work unrelated to the contributions.
> > > >
> > > > Even though looking at the image may not fully solve this issue, it may, even significantly, mitigate the issue caused by being completely agnoistic to the actual asset appearance, e.g., the shape of the bathhub.
> > > >
> > > > At last, I turn to agree that the refiner does not mean a fundamental issue, but like a workaround on a powerful but capability-constrainted method. I acknowledge this concern addressed, and still welcome further discussions or clarifications on the LLM evaluators.

---

> > > > > ### Author Response · Authors · 2025-08-09
> > > > >
> > > > > Thank you for your continued feedback.
> > > > >
> > > > > I agree that our method cannot guarantee that the LLM will always make fair and consistent judgments across different cases. However, to the best of our knowledge, methods using LLMs for evaluation all suffer from this issue to some extent, and it is also difficult to design a metric that can reliably verify whether the LLM has applied a consistent rubric across scenarios. In our practice, we have already refined the evaluation rubric as much as possible to obtain more reliable feedback. Specifically, the effectiveness of our reward can be seen by comparing it with the “simple reward” in our ablation study (the prompt is also provided in Appendix A.1). The simple reward asks the LLM to directly score based on physical plausibility, semantic consistency, and instruction-following, which is closer to “letting the AI give an answer at first glance.” Our method achieves significantly better performance than the simple reward across all metrics. In addition, this issue does not affect our core contribution. The core contribution of our method lies in the pathway of bypassing constrained optimization and enabling more flexible open-vocabulary 3D scene synthesis with limited data, and its effectiveness has been demonstrated in user study and comparative experiments.

---

### Official Review · Reviewer_y4q3 · 2025-07-02

**Clarity:** 2
**Significance:** 2
**Originality:** 2
**Rating:** 4
**Confidence:** 4

**Summary:**

The authors propose a new system for open-vocab, languaged-conditioned, indoor layout synthesis. The system uses LLMs/VLMs to (1) iterate on a language specification of a birdseye view floorplan, then (2) convert 2d plans to likely 3d object heights/specs and finally (3) refining object locations. This is comprised of structured prompts for eliciting CoT, describing BEV layouts, and LLM-reward for DPO finetuning. The provided qualitative results are strong, and the method's results are preferred in LLM-based quantitative evaluations.

**Questions:**

How were the prompts of Figure 3 selected? Are the results shown random samples (e.g. with a pre-determined seed = 0 for each method, best-of-1)? Are the Fig3 results representative in quality of what I might see if I ran your method for seeds 0..5 on identical prompts as prior work? (e.g. the starwars office prompt of Holodeck Fig 1)

How can we be assured of the usefulness of larger samples of scenes for downstream use (by humans, in simulators or other graphics) ? Please comment on the validity of quantitative results (see above). I estimate it will difficult to address without a tightly controlled user study (randomized, >7 participants, well crafted survey) which is beyond the scope of a rebuttal.

What is the mean/min/max CPU runtime, API credit usage, and/or GPU of the method under various settings? Is this new cost-performance tradeoff appealing for downstream users?

**Ethical Concerns:**

["NO or VERY MINOR ethics concerns only"]

**Final Justification:**

My original rating was based primarily on concerns or misunderstandings on evaluation, which the authors have fully addressed in rebuttal. The authors' use of VLM-graded perceptual studies is in line with prior work, they added a user study, additional baseline systems, and provided the requested runtime / cost averages.

I share other reviewers' view that the method is not hugely insightful or technically interesting. The authors state their novelty lies in "the new perspective [CoT/DPO] enable for 3D scene layout generation", however the transition into less structured and lower level language primitives strikes me as more gradual over the course of multiple prior works e.g. Holodeck. The output here remains somewhat structured, and feedback still comes from a geometric checker.

However, my explicit concerns have been largely addressed, and the work's favourable evaluation results mean it may be useful for directly generating 3D scenes for robotics, and as an exemplar of maximally-LLM-based arrangement as a comparison for future work.

**Limitations:**

yes

**Paper Formatting Concerns:**

none, minor typos only.

**Quality:**

3

**Strengths And Weaknesses:**

qualitative results are strong but not plentiful or controlled:
- strength: the model produces very good layouts with good instruction following, especially in the kitchen and home-theater examples - these feature objects not attached to walls which are minimally constrained and difficult to place correctly.
- weakness: needs larger & tightly controlled qualitative results
	- only one example is shown for each granularity of prompt. we do not know the variance / stability of the method when repeatedly sampled with the same prompt.
	- not stated how the prompts were selected - are they representative/random/standardized, or designed to showcase the strength of the authors method?
	- ATISS & Infinigen provide large samples of 25+ _randomly selected_ qualitative results in their supplements, this work does not.

quantitative results have two major flaws:
- Authors follow LayoutGPT in using metrics which are _evaluated by GPT4o_ (with no human study) which I believe is a major flaw in evaluation of this work.
	- using GPT-4o as the validation score, when iterating against GPT-4o's criticisms is an integral part of the method, is invalid. This is akin to optimizing against the validation set - if we can see (somewhat stochastic samples of) the final validation metric and change our answer, it is no longer a validation metric.
	- I recognize LLMs-as-judges is a widely used paradigm in NLP, but I believe it is invalid for layout as VLMs are not known to possess strong spatial reasoning without significant scaffolding / extra engineering (as evidenced by goals of this work)
	- Every baseline method in table 1 provides a user study (Holodeck, I-Design, LayoutGPT). The authors do not perform a user study, they instead cite that LLMs have correlated with users in the past. But this is questionable since the authors method is different & more LLM/VLM reliant than prior methods.
	- Out-of-bounds/Collision metrics are quantitative but not sufficiently informative. Objects could topple when settled in simulation, or have their affordances violated (objects blocking chairs/ovens) so these are useful to include but not sufficient as a sole evaluation.
- Table 1 omits comparison with ATISS, which is well known and which I consider closely related in method. Less critically, the authors could also evaluate DiffuScene.

Discussion of the method's cost is inadequate.
- only mention is "introduce additional inference time" L315
- The authors should know the average spend on OpenAI APIs per scene, but the cost is not stated.
- Tab1 should list costs of each method. Fig.3 could report per-scene costs (is the cost higher for more complex scenes? this would be interesting). See questions below.
- If the cost is high, is the method better than baselines on an equal-cost basis (baselines run for longer, or authors method is allowed fewer iterations?)
- The authors fail to argue that the performance-cost tradeoff of adopting their method is worthwhile for downstream tasks. Would roboticist use the method if scalability costs $X/scene in API credits/GPU-time? We cannot know as the cost is not stated.
- Increased cost would be a predictable downside of their method of replacing constraint solvers with LLM reasoning, which is the core proposal. One gains flexiblity / open-vocab, but loses the computational efficiency of a constraint solver.

Distinction from prior work, especially related to "constraints", is imprecise:
- "mainly focus on modelling the distribution of indoor scene layouts, wheras our method leverages limited data to learn the underlying placement logic of indoor layouts" (L85) - what is the difference getwen these two? is "learning the logic from data" not an example of learning the distribution?
- "explicit logical outputs" L101 instead of constraints - the methods outputs (L155) are very similar to constraints but are "solved" by an LLM
- "predefined constraints"... "inherently sacrifice flexibility" L36 - not necessarily - if the predefined DSL is sufficiently low-level (has primitives that are universal to all scenes, not specific to objects), they may apply to many/all scenes

The authors presuppose many ambitious properties of LLMs/VLMs e.g " general reasoning capabilites" (L88). these are still open areas of research and evaluation - LLM reasoning can fail on examples that are trivial for humans.

Quality Justification (technically sound?): lacks valid evaluation results, lacks sufficient qualitative results, lacks any discussion of cost.

Clarity Justification (clearly written?) 3. fairly clear, especially once one reads the prompts in the supplement. short excerpts of these could be moved to the main paper.

Significance Justification (impactful for community? likely to build on ideas?): Generating useful simulation environments for robotics is potentially very significant. But it is impossible to predict adoption given unknown cost.

Originality (new insights?):  CoT or DPO are now standard practice in using LLMs, I do not find the nature of their use in this paper insightful. The novelty of the work is a set of structured prompts for layout specification, eliciting standard LLM chain-of-thoughts, and/or rewards. LLMs with structured outputs, that are prompted to perform CoT, and/or rewards from another LLM prompt, are prevalent.

---

> ### Author Rebuttal · Authors · 2025-07-31
>
> Thank you sincerely for your valuable time and insightful comments.
>
> ### **W1 & Q1: Qualitative results**
>
> We provide multiple examples per granularity in the main paper and supplementary materials, though only one visualization per prompt is shown. Due to rebuttal formatting limits, more visuals cannot be included here. Instead, we present a quantitative study measuring the standard deviation of our model over ten repeated generations using the same prompt or scene type. Each table row shows results from one randomly sampled prompt under the given method.
>
> | Model    |   Pos |   Rot |   PSA |
> |:---------|----------:|----------:|----------:|
> | Ours (fine)     |      5.83 |      5.48 |      6.78 |
> | Ours (medium)   |      4    |      6.78 |      6.32 |
> | Ours (coarse)   |      5.83 |      2.93 |      4    |
> | ATISS    |      2    |      4.9  |      7.8  |
> | Holodeck |      4    |      7.28 |      6.75 |
>
> As noted in the Appendix A.3, we generated 45 scenes per granularity. For I-Design that sometimes fails to produce valid results, we exclude those cases. Prompts in Figure 3 are randomly chosen from successful generations. We include 10+ randomly selected qualitative results in the paper and supplement, and will add more examples in a revision.
>
> ### **W2 & Q2: GPT-4o based evaluation is flawed; no human study**
>
> We adopt the GPT-based metric from LayoutVLM (CVPR 2025), which has been validated to align well with human judgments. While VLMs have limits in fine-grained spatial reasoning, we distinguish between generative spatial reasoning (precise 3D placement) and visual content understanding. The former requires exact spatial computation, the latter image perception. Therefore, properly calibrated VLM-based evaluation like LayoutVLM remains a valuable reference for layout quality.
>
> In our pipeline, this metric is used only for evaluation, avoiding overfitting since training rewards come from detecting objective errors (e.g., collisions, CoT mismatch), which differ from the evaluation metric. We follow prior works like LayoutGPT and LayoutVLM using these to provide extra information. To further validate, we conduct a user study with 25 compensated volunteers experienced in computer science and 3D vision, who score layouts from various methods on three criteria: Semantic Compliance (alignment with textual description), Layout Rationality (plausibility of object placement), Physical Plausibility (realism of object support, no penetration, etc.).
>
> Each method’s results were randomly ordered and anonymized. The average scores are reported below:
>
> | Method      | Level   | SemComp | PhyPlaus | LayRatl |
> |-------------|---------|---------|----------|---------|
> | LayoutGPT   | coarse  | 3.72    | 1.84     | 1.72    |
> |             | medium  | 2.88    | 1.64     | 1.76    |
> |             | fine    | 3.00    | 2.32     | 2.04    |
> |             | overall | 3.20    | 1.93     | 1.84    |
> | I-Design    | coarse  | 3.72    | 2.72     | 2.44    |
> |             | medium  | 3.08    | 3.36     | 2.48    |
> |             | fine    | 3.32    | 3.72     | 2.72    |
> |             | overall | 3.37    | 3.27     | 2.55    |
> | Holodeck    | coarse  | 4.12    | 3.64     | 2.84    |
> |             | medium  | 2.80    | 2.52     | 2.48    |
> |             | fine    | 2.40    | 2.60     | 2.04    |
> |             | overall | 3.11    | 2.92     | 2.45    |
> | Ours        | coarse  | 4.56    | 4.64     | 4.52    |
> |             | medium  | 4.60    | 4.68     | 4.64    |
> |             | fine    | 4.84    | 4.72     | 4.80    |
> |             | overall | 4.67    | 4.68     | 4.65    |
>
> We also include comparison with ATISS, DiffuScene, and Infinigen:
>
> | Method     | PhyPlaus |  LayRatl |
> |:-----------|----------|----------|
> | ATISS      | 3.66     | 3.54     |
> | Infinigen  | 4.66     | 4.44     |
> | Diffuscene | 3.46     | 3.1      |
> | Ours       | 4.62     | 4.48     |
>
> While they perform strongly overall, our method matches or surpasses their performance on certain scene categories, and additionally supports open-vocabulary generation.
>
> ### **W3: Table 1 omits ATISS and DiffuScene**
>
> ATISS and DiffuScene are not directly comparable to our method due to fundamental differences in input and scope: 1. Our method supports open-vocabulary text inputs, while ATISS and DiffuScene are trained to generate specific scene types (e.g., bedrooms or living rooms). 2. Our model is LLM-based, while theirs are layout diffusion or scene synthesis networks that do not mainly rely on natural language inputs.
>
> Despite this, to facilitate a fairer comparison, we randomly generate 20 samples for bedrooms and living rooms and evaluated across standard metrics:
>
> | Method     | Scene      |   Pos |   Rot |   PSA |
> |:-----------|:-----------|------:|------:|------:|
> | ATISS      | bedroom    | 81    | 70    |    86 |
> |            | livingroom | 71.5  | 79.5  |    76 |
> |            | overall    | 76.25 | 74.75 |    81 |
> | Diffuscene | bedroom    | 66    | 63    |    70 |
> |            | livingroom | 66.5  | 66    |    60 |
> |            | overall    | 66.25 | 64.5  |    65 |
> | Ours       | bedroom    | 82.5  | 75    |    96 |
> |            | livingroom | 86    | 75.4  |    80 |
> |            | overall    | 84.25 | 75.2  |    88 |
>
>
> ### **W4 & Q3: Method cost**
> We discuss the cost here and will add more discussions in the revised paper.
>
> Monetary cost:
> We don’t report per-scene monetary costs due to inconsistent baselines using different models (e.g., GPT-3.5, GPT-4, GPT-4o) with varying API pricing, which makes fair comparison difficult. To ensure transparency, we release a fully open-source pipeline that can run locally or on open LLMs, where cost-free variants outperform original baselines on layout quality. If needed, we can run experiments with a unified model (e.g., GPT-4o) to report average per-scene costs.
>
> Inference time cost:
> We evaluate inference time systematically by measuring average, minimum, and maximum times for generating layouts from 10 random prompts at three granularity levels, all on the same platform and network. Except for LayoutGPT, which is limited to specific scene types, our method achieves better results with comparable inference times (Unit: seconds):
>
> | Method                      |   Average |   Min |   Max |
> |:----------------------------|---------------------------:|-----------------------:|-----------------------:|
> | LayoutGPT                   |                       5.87 |                   5.32 |                   6.67 |
> | IDesign (~60% failure rate) |                     166.86 |                 148.12 |                 229.74 |
> | Holodeck                    |                     210.23 |                 172.59 |                 291.88 |
> | Ours                        |                     185.24 |                 109.45 |                 298.13 |
>
>
> ### **W5: Ambiguity in constraint-related distinction**
> We will clarify the details in the revised version.
>
> Prior learning-based methods directly learn distributions of numerical layouts. Our method instead learns the logic behind object placement, which generalizes better across unseen scenes. For example, ATISS, LayoutGPT has difficulty generating scenes with open vocabulary.
>
> Predefined constraints can theoretically generalize, but manually defining them is costly and often fails across diverse scene types. Prior work mainly uses coarse spatial relations like relative directions or distances. While LLMs handle such constraints more easily than generating exact layouts, this limits realism and diversity and struggles with fine-grained descriptions. For example, “the cake is placed at the top-left corner of the table” cannot be captured by simple rules like “A is left/right/above/below B” or “A is X meters from B.” In our method, such detailed spatial relations are treated as constraints generated by the LLM. User study show clear advantages in handling these fine-grained descriptions.
>
> ### **W6: Originality**
>
> Our novelty lies not in the use of CoT or DPO themselves, but in the new perspective they enable for 3D scene layout generation. Prior methods (e.g., ATISS) rely on fully supervised training over limited datasets, learning fixed spatial distributions that struggle to generalize to new categories or diverse scenes. Recent efforts (e.g., Holodeck, I-Design) have begun using LLMs to bridge this gap, but these typically depend on manually defined spatial constraints and external solvers, limiting flexibility and fine-grained spatial control. A common consensus regarding this type of work is that LLM finds it difficult to generate precise numerical layouts. In contrast, our method uses CoT and DPO to enhance generalizable reasoning, enabling LLMs to generate complete numerical layouts directly and opening up new ways of thinking.
>
> We also introduce an entropy weight–based reward integration scheme to balance multiple layout quality aspects. While prior LLM research mainly targets math or coding, 3D spatial reasoning offers a valuable testbed. We provide a feasible method for LLM to handle generative 3D spatial reasoning problems. We will open source the CoT dataset and code to support further progress in this area.
>
> ### **W7: Ambitious Properties**
>
> We apologize for the wording that may have caused misunderstanding. While LLMs are not flawless, our work only leverages their commonsense reasoning for spatial relation generation, which is a narrower and more stable use case supported by prior work and empirical results. We will use more precise wording to reduce misunderstandings.
>
> We hope our responses can address your concerns. Please feel free to inform us of any further issues.

---

> > ### Comment · Reviewer_y4q3 · 2025-08-04
> >
> > Thanks for your response.
> >
> > I think it would be useful to state the API cost of the method. I agree that rerunning all prior work using GPT4o would be overly burdensome. However, this does not prevent the authors from stating the model & api cost of just their own method as it stands today. For example, Holodeck states that their method consumes $0.20 of API credits using GPT-4-1106-preview. Even though they do not show API costs for other methods, knowing the cost of their method is still useful. The authors should be able to state a similar approximate average cost without running any new experiments.
> >
> > I would also hesitate to call the local/open-source LLM methods "cost free", since these would now just have a GPU cost, and GPU runtime is not stated.
> >
> > Regarding the prompts in Fig 3 I had hoped to know how the underlying prompts were determined for all experiments. The sentence "prompts in Figure 3 are randomly chosen from successful generations" to me means that there is an initial set of prompts X, which are used as input to the system, and a random subset X' succeed and are shown in Fig 3. But this does not mean that X is random, and does not say how it was selected. The best evaluation would be if X was somehow random or derived from another dataset, although I recognize this is may not be done in prior work.

---

> > > ### Author Response · Authors · 2025-08-05
> > >
> > > Thank you for your continued feedback.
> > >
> > > We now report the average monetary cost of our method on the test set, which is an average of \$0.0611 per scene using OpenAI API. This is based on our current setup described in the paper: GPT-4o is used as both the 3D Layout Generator and the Spatial Evaluator, while o1 is used as the Quantitative Evaluator. Importantly, this cost reflects a conservative upper bound, as we did not use batch API calls. In practice, for large-scale or non-interactive downstream applications, batching can further reduce the cost by approximately 50%. Considering both the time efficiency and the monetary cost, we believe our method strikes a favorable balance between performance and scalability, especially given its flexibility and generalization capabilities beyond what traditional solvers offer.
> > >
> > > Regarding the initial prompt set X, we clarify that it was generated systematically. The scene categories and corresponding descriptions were generated automatically by the LLM in response to our instruction. Specifically, we instructed the model to generate scene descriptions for a diverse set of indoor environments while excluding common types (e.g., bedroom, living room, dining room, study), and to cover three levels of granularity (coarse, medium, fine-grained) for each category, as detailed in the prompt above (same as shown in Appendix A.1):
> > > ```
> > > You are asked to generate indoor scene descriptions at three levels of granularity: coarse, medium, and fine-grained. Please follow the instructions carefully.
> > > There are three types of granularity:
> > > 1. Coarse: List the main objects in the room without mentioning where they are.
> > >    Example: "A home gym with a treadmill, yoga mat, dumbbell rack, water dispenser, and a large mirror."
> > > 2. Medium: Describe the approximate spatial relationships between major object groups.
> > >    Example: "In a playroom, a toy shelf stands against the right wall, a bean bag lies in the corner near the window, and a round play mat is placed in the center."
> > > 3. Fine-grained: Provide precise, detailed spatial arrangements among individual objects.
> > >    Example: "A small square table is placed in the center of the room. On the front right corner of the table sits a red toolbox, with a measuring tape coiled beside it. A yellow stool is tucked in on the left side, and a desk lamp stands at the rear center of the table."
> > >
> > > You will generate scene descriptions for {num_scene_types} different indoor scene categories, excluding common categories such as bedroom, living room, dining room, and study.
> > > For each scene category, generate:
> > > {num_coarse_per_type} descriptions at coarse granularity,
> > > {num_medium_per_type} at medium granularity,
> > > {num_fine_per_type} at fine-grained granularity.
> > > Each scene description should be accompanied by room dimensions: length, width, and height (each must be an integer <= 256).
> > >
> > > Output the results in a strict json format as follows:
> > > [
> > >   {
> > >     "scene_type": "laundry room",
> > >     "granularity": "coarse",
> > >     "description": "A laundry room with a washing machine, dryer, laundry basket, shelves, and detergent bottles.",
> > >     "room_size": {
> > >       "length": 256,
> > >       "width": 171,
> > >       "height": 240
> > >     }
> > >   },
> > >   ...
> > > ]
> > > ```
> > >
> > > Please let us know if there are any remaining concerns. We'd be happy to clarify further.

---

### Decision · Program_Chairs · 2025-09-17

**Decision:**

Accept (poster)

**Comment:**

This paper introduces DirectLayout, a novel framework for generating 3D indoor scene layouts directly from textual descriptions. The core idea is to bypass the common paradigm of generating intermediate representations (like scene graphs) followed by constrained optimization. Instead, the authors propose a method where a Large Language Model (LLM) is trained to directly output numerical layout parameters (e.g., coordinates, dimensions, orientation). The framework decomposes the task into three stages: generating a 2D Bird's-Eye View (BEV) layout, lifting it to 3D, and an iterative refinement step to align the layout with the generated 3D assets.

Initial reviews were mixed, but following the rebuttal and subsequent discussion, a clear consensus emerged. Two reviewers are solid "Accepts," and one "Borderline Accept" reviewer stated that their major concerns were "fully addressed." Only one reviewer remains at "Borderline Reject," still having valid reservations about the technical novelty and the ultimate reliability of LLM evaluators.  However, even this reviewer acknowledged that many of their concerns were addressed.

Given these points above, we have decided to accept the paper.